

# Regional temperature change potentials for short lived climate forcers from multiple models

Borgar Aamaas[1], Terje K. Berntsen[1,2], Jan S. Fuglestvedt[1], Keith P. Shine[3], William J. Collins[3]

[1]CICERO Center for International Climate Research, PB 1129 Blindern, 0318 Oslo, Norway

[2]Department of Geosciences, University of Oslo, Norway

[3]Department of Meteorology, University of Reading, Reading RG6 6BB, United Kingdom

*Correspondence to*: Borgar Aamaas (borgar.aamaas@cicero.oslo.no)

**Abstract.** We calculate the absolute regional temperature change potential (ARTP) of various short lived climate forcers (SLCFs) based on detailed radiative forcing (RF) calculations from four different models. The temperature response has been estimated for four latitude bands (90-28° S, 28° S-28° N, 28-60° N, and 60-90° N). The regional pattern in climate response not only depends on the relationship between RF and surface temperature, but also on where and when emissions occurred and atmospheric transport, chemistry, interaction with clouds, and deposition. We present four emissions cases covering Europe, East Asia, the global shipping sector, and the globe. Our study is the first to estimate ARTP values for emissions during Northern Hemisphere summer (May-October) and winter season (November-April). The species studied are aerosols and aerosol precursors (black carbon (BC), organic carbon (OC), $SO_2$, $NH_3$), ozone precursors ($NO_x$, CO, volatile organic compound (VOC)), and methane ($CH_4$). For the response to BC in the Arctic, we take into account the vertical structure of the RF in the atmosphere, and an enhanced climate efficacy for BC deposition on snow. Of all SLCFs, BC is the most sensitive to where and when the emissions occur, as well as giving the largest difference in response between the latitude bands. The temperature response in the Arctic is almost 4 times larger and more than 2 times larger than the global average for Northern Hemisphere winter emissions for Europe and East Asia, respectively. The latitudinal breakdown gives likely a better estimate of the global temperature response as it accounts for varying efficacies with latitude. An annual pulse of non-methane SLCFs emissions globally (representative of 2008) leads to a global cooling. Whereas, winter emissions in Europe and East Asia give a net warming in the Arctic due to significant warming from BC deposition on snow.

## 1 Introduction

Climate is influenced by a multitude of emissions with varying impacts (e.g., Myhre et al., 2013). Emissions of short lived climate forcers (SLCFs), such as black carbon (BC), organic carbon (OC), $SO_2$, $NH_3$, $NO_x$, CO, and volatile organic compounds (VOCs), affect the composition of the atmosphere primarily on time scales of days to a few months. $CH_4$ is often also included because its lifetime of around 10 years is shorter than or comparable to climate response timescales. The temporal variation in the geographical pattern of SLCF emissions has changed over time, with emissions typically being high in the early phases of industrialization, and then gradually being reduced due to air quality concerns and technological improvements. Nevertheless, emissions are still growing in many parts of the world, and there is a growing focus politically to develop mitigation strategy for the SLCFs to





achieve both improved air quality and slowing global warming (Schmale et al., 2014;Shindell et al., 2012;Stohl et
al., 2015).
Due to the short atmospheric lifetimes, emissions of SLCFs lead to a spatial pattern in radiative forcing (RF) that
is more inhomogeneous than for emissions of long-lived greenhouse gases such as $CO_2$. It is well established that
there is not a close relationship between the RF pattern and the surface temperature response pattern, due to
modifications by heat transport in the atmosphere and ocean and the spatial variability in climate feedbacks (e.g.,
Boer and Yu, 2003). However, as shown by Shindell and Faluvegi (2009) and Shindell (2012), it is possible to
establish relationships between the RF pattern caused by a certain component and the response in broad latitude
bands.  Recently, Najafi et al. (2015), have shown from observational and model data that there is a distinct
difference in the Arctic response to the overall forcing by ozone, aerosols and land-use, compared to other latitude
bands.
Emission metrics are simple tools based on comprehensive model simulations that relate emissions to a certain
response (physical climate change or economic damage), e.g. Fuglestvedt et al. (2003);Tol et al. (2012). The most
widely used emission metric, the Global Warming Potential (GWP), is given by the integrated RF (over a time
horizon of H years) in response to a pulse emission. Shine et al. (2005) introduced the Global Temperature change
Potential (GTP), using the surface temperature change (after a time horizon of H years) for the response. Emissions
metrics have typically estimated a global effect due to global emissions (e.g., Aamaas et al., 2013). A first step
going beyond global means was to quantify the global response based on regional emissions for SLCFs (Berntsen
et al., 2005;Wild et al., 2001;Stevenson et al., 2004;Fuglestvedt et al., 2010;Collins et al., 2013;Aamaas et al.,
2016;Fry et al., 2012). By introducing the concept of regional temperature potentials (RTP), Shindell and Faluvegi
(2010) extended the metric concept to include regional responses (in terms of surface temperature change in broad
latitude bands) to regional emissions.
In addition to the regionality, the timing of the SLCFs emissions matter. This is potentially important since the
photochemistry in the atmosphere, lifetime, atmospheric transport and forcing efficiency is likely to vary between
the seasons. As some sources (e.g. domestic heating and agricultural waste burning) have a large seasonal cycle,
using seasonal RTP metrics might have a significant impact on the evaluation of cost-effectiveness of mitigation
measures.
Here we use detailed multimodel calculations of the relationship between emission location and the resulting
specific RF (RF per Tg/yr emissions) for SLCFs (Bellouin et al., 2016) (Sect. 2.1) and the regional climate
sensitivities (e.g., Shindell and Faluvegi, 2009) to estimate ARTPs for a range of aerosols, aerosol precursors, and
ozone precursors (BC, OC, $SO_2$, $NH_3$, $NO_x$, CO, and VOC), and $CH_4$ (Sect. 2.2). While some of our findings
confirm the results by Collins et al. (2013), our analysis build on that work in several ways (see also Aamaas et
al., 2016). Our study is the first to calculate ARTPs for $NH_3$ emissions. The treatment of BC in the Arctic is more
complex which has a high influence on the ARTPs for BC. Aspects of the aerosol effects on ozone precursors are
also novel. For the first time, we distinguish between ARTPs for emissions taking place during Northern
Hemisphere (NH) summer (May-October) and winter (November-April). ARTP metrics are calculated for regional
emissions from Europe, East Asia and the shipping sector, as well as global emissions (Sect. 3.1). The ARTP





values are applied to global emissions in Sect. 3.2. We also make a comparison of ARTPs with AGTPs (Sect. 3.3).
Uncertainties are discussed in Sect. 3.4, and we conclude in Sect. 4.

## 2    Material and methods

### 2.1    Radiative forcing

The RFs that are the basis for the ARTP calculations of the SLCFs are calculated using 4 different chemistry
climate models or chemical-transport models presented by Bellouin et al. (2016); see details about the models in
Table 1. RFs are produced based on a control simulation and numerous perturbation simulations that consider a
20% emission reduction in one type of species and one region in NH summer or winter. The ECLIPSE emission
dataset applied here was created with the GAINS (Greenhouse gas-Air pollution Interactions and Synergies) model,
see Stohl et al. (2015). The regional pattern of the RFs is taken into consideration with four latitude bands, southern
mid-high latitudes (90-28° S), the Tropics (28° S-28° N), northern mid-latitudes (28-60° N), and the Arctic (60-
90° N), as forcing-response coefficients are only available for those latitude bands in the literature (e.g., Shindell
and Faluvegi, 2010;Shindell, 2012).
We compute ARTPs for six different processes that contribute to the RF for each species (see Fig. 1 for details).
The quantification of these processes are given by the RF data from Bellouin et al. (2016). For the general
circulation models, the RFs of the aerosol perturbations are calculated online using two calls to the radiation
scheme. This method involves diagnosing radiative fluxes with and without the perturbation. These RFs do not
include rapid adjustments (even in the stratosphere). For the OsloCTM2 chemistry transport model and the RF
exerted by the ozone precursors in all the models, RF is computed by offline radiative transfer codes. The RF for
methane is based on the analytical expression that includes stratospheric adjustments (Myhre et al., 1998), which
gives a global mean. Based on this global RF estimate, we apply the latitudinal pattern in RF for methane and
methane induced ozone response in Collins et al. (2013). This pattern is based on an ensemble of 11 global
chemical transport models that evaluated a global reduction of $CH_4$ mixing ratio, where RF was calculated using
the method developed by the NOAA Geophysical Fluid Dynamics Laboratory (Fry et al., 2012).
For aerosols and aerosol precursors, all four models calculate the aerosol direct and 1[st] indirect (cloud-albedo)
effect, except ECHAM6 which only includes direct RF. In this study, we group together the aerosol direct and 1[st]
indirect (cloud-albedo) effect and name this process *aerosol effects*. In addition, OsloCTM2 estimated the RF from
BC deposition on snow and the semi-direct effect. The semi-direct effect is quantified in Bellouin et al. (2016) by
prescribing control and perturbed distributions of BC mass-mixing ratios in the CAM4 model and using 30-year
simulations with fixed sea-surface temperatures to suppress the long-term response. For the ozone precursors and
$CH_4$, the total RF takes into account the aerosol direct and 1[st] indirect effects, short-lived ozone effect, methane
effect, and methane-induced ozone effect. Only OsloCTM2 includes an estimate for nitrate aerosols, which is
added to the *aerosol effect* quantification in the other models.
The best estimate of a species' RF has been calculated as the sum of all processes, in which the average across the
models is used for each process. Not all models have estimated RFs for all species and processes. In addition,
ECHAM6 is excluded in the best estimate for BC, OC, and $SO_2$, since it did not estimate the 1[st] indirect effect. For
BC deposition on snow, the BC semi-direct effect, and nitrate aerosol, the best estimate is solely based on the





OsloCTM2 model, while the best estimate are based on three models for all other processes (*aerosol effects*, short-
lived ozone, methane, and methane-induced ozone).
For the high and low estimates of RF for each emission case, we find these values by taking the sum of the highest
and lowest values, respectively, from all models for each individual process.
The emission regions are defined according to tier1 Hemispheric Transport of Air Pollution (HTAP) regions (see
Bellouin et al., 2016). Europe is defined as Western and Eastern Europe up to 66°N including Turkey. East Asia
includes China, Korea, and Japan. Shipping is the global shipping sector. The global emissions category excludes
this shipping activity. As RF values are also available for the remaining land areas outside of Europe and East
Asia, results from the rest of the World are presented in SI Sect. 2.
**2.2  Regional temperature change potentials**
The regional temperature response has been calculated on the basis of RF in the latitude bands and regional climate
sensitivities, as well as the temporal evolution of an idealized temperature response. Even though our estimates
are based on seasonal emissions, the temperature responses calculated are annual means. The general expression
for the ARTP following a pulse emission of component $i$ ($E_i$) in region $r$ which  leads to a response in latitude
band $m$ is (e.g., Collins et al., 2013):
$$ARTP_{i,r,m,s}(H) = \sum_l \int_0^H \frac{F_{l,i,r,s}(t)}{E_{i,r,s}} \times RCS_{i,l,m} \times R_T(H-t)dt \qquad (1)$$
$F_{l,r,s}(t)$ is the RF in latitude band $l$ due to  emission in region $r$ in season $s$ as a function of time ($t$) after the pulse
emission $E_{r,s}$ (in Tg). The $RCS_{i,l,m}$ is a matrix of regional response coefficients based on the RTP concept (unitless,
cf. Collins et al., 2013). As these response coefficients are here normalized, they contain no information on climate
sensitivity, only the relative regional response pattern. The global climate sensitivity is included in the impulse
response function $R_T$, which is a temporal temperature response to an instantaneous unit pulse of RF (in K/(Wm$^{-}$
$^2$)). We assume that the time evolution of temperature in each response band follows the global-mean time
evolution.  We base our temperature response on that of the HadCM3 climate model (Boucher and Reddy, 2008)
with an equilibrium climate sensitivity of 1.06 K/(W m$^{-2}$), which translates to a 3.9 K warming for a doubling of
$CO_2$ concentration. This is the same climate sensitivity as for our absolute Global Temperature change Potential
(AGTP) calculations on the same RF dataset (Aamaas et al., 2016).
Regional temperature responses of an emission scenario $E(t)$ can be calculated with these ARTP values by a
convolution (see also Aamaas et al., 2016). The temperature response is:
$$\Delta T_i(t) = \int_0^t E_i(t') \times ARTP_i(t-t')dt' \qquad (2)$$
**2.2.1    For species with lifetimes less than one year**
For SLCFs with atmospheric lifetimes (or indirect effects causing RF) much shorter than both the time horizon of
the ARTP and the response time of the climate system (given by the time constants in $R_T$ above), the general
expression for the ARTP can be simplified to:
$$ARTP_{i,r,m,s}(H) = \sum_l \frac{F_{l,i,r,s}}{E_{i,r,s}} \times RCS_{i,l,m} \times R_T(H) \qquad (3)$$





$F_{i,r,s}$ is the RF over a year where emissions of component $i$ ($E_{i,r}$,s in Tg/yr) in emission region $r$ occur during season
$s$, either during NH summer or winter.

### 2.2.2    For species that affect methane

Methane has an adjustment time comparable to the time horizon of the ARTP and the response time of the climate
system. So, for species that affect methane, an additional impulse response function that describes the atmospheric
decay of methane must be included ($R_F$). In this case, we add such a function, which governs the methane and
methane-induced ozone effects for the ozone precursors (NOx, CO, and VOC) and $CH_4$.
$$R_F(t) = e^{-t/\tau}, \tag{4}$$

where $\tau$=9.7 yr is the average adjustment time for methane in the three models. For these species, this additional
temperature perturbation due to these processes has to be included:
$$ARTP(R_F\ response)_{i,r,m,s}(H) = \sum_l \int_o^t \frac{F_{l,i,r,s}}{E_{i,r,s}} \times R_F(H-t) \times RCS_{i,l,m} \times R_T(H-t)dt \tag{5}$$

### 2.2.3    Forcing-response coefficients

The unitless regional sensitivity matrix ($RCS_{i,l,m}$) is estimated based on literature values of regional response
coefficients in K/(W m$^{-2}$) (see Sect. 1 in Supporting Information for tabulated coefficients). All these response
coefficients from the different literature sources have been normalized to the global response in those studies.
While the specific regional response coefficients have been estimated in other studies based on climate sensitivities,
the normalization to the global response removes the implicit climate sensitivities in the RCS values. We do several
adjustments and refinements of the RCS values (see this section and Sect. 2.2.4); in each case, we normalize the
response coefficients and make sure that the climate sensitivity in our ARTP calculations are only incorporated in
$R_T$.
As such, RCS matrices only exist for annual emissions, we assume we can apply the same set of matrices for
emissions during NH summer and winter. For the scattering aerosols and aerosol precursors ($SO_2$, OC, $NH_3$), we
use the coefficients tabulated in Shindell and Faluvegi (2010), which are the mean responses of $CO_2$ and $SO_2$. The
same values are used for the long-lived effects (methane and methane-induced ozone) of the ozone precursors and
$CH_4$. For the short lived effects of the ozone precursors and $CH_4$, we apply the $O_3$ coefficients in Shindell and
Faluvegi (2010) as tabulated in Collins et al. (2013).
For BC, the regional sensitivity matrix is based on several sources, and the details for the Arctic-to-Arctic
responses are described in Sect. 2.2.4. For other latitude bands, the matrix for the *aerosol effects* is given by BC
forcing-response coefficients from Shindell and Faluvegi (2009) as tabulated in Table 3 in Collins et al. (2013)
and the matrix for the semi-direct effect is from the $CO_2$ coefficients shown in Shindell and Faluvegi (2010) based
on Shindell and Faluvegi (2009). The semi-direct effect can potentially be included either in the response based
on RCS values or in the RF. Our approach is to include the semi-direct effect in the RF and not in the RCS values,
see next paragraph for details. The relationship for the deposition of BC on snow is also given by the $CO_2$
coefficients shown in Shindell and Faluvegi (2010). For the snow albedo effect, we have assumed an efficacy of
3 for all RF occurring outside of the Arctic (Myhre et al., 2013).



Our method differs from Shindell and Faluvegi (2009) as we have calculated the semi-direct effect independently.
Since Shindell and Faluvegi (2009) did not have any rapid adjustments in their sensitivities on RFs, the rapid
adjustments are implicitly included in their sensitivity coefficients. The reason is that in the GCM simulations used
to calculate the forcing-response coefficients (Shindell and Faluvegi, 2009;Flanner, 2013), semi-direct effects are
treated as feedbacks and as such they are included in the forcing-response coefficients. When we normalize to the
global response to find the RCS coefficients, we normalize on the global $CO_2$ response given by Shindell and
Faluvegi (2009) for all the species to avoid double counting.

### 2.2.4 Refinement of Arctic Response to BC

We do two refinements of the forcing-response coefficients for RFs occurring in the Arctic, one for the *aerosol*
*effects* in the atmosphere and one for the effects due to BC on snow. We first discuss how we handle the *aerosol*
*effects* in the atmosphere.
For BC in the Arctic, the forcing by absorption takes place in a generally stably stratified atmosphere. The transport
of BC to the Arctic occurs approximately along isentropic surfaces; thus emissions from East Asia are generally
at a higher altitude than emissions from Europe. The BC particles cause also dimming at the surface. In the Arctic,
heat is not easily mixed down to the surface. The efficacy of BC forcing depends highly on the altitude of the BC
(Flanner, 2013;Lund et al., 2014;Sand et al., 2013). To account for this the RTP concept is modified for BC forcing
in the Arctic. The contribution by RF exerted in the three latitude bands outside the Arctic to Arctic warming
($ARTP(ex\text{-}Arc)_{BC,r,Arc,s}$) is calculated with the standard method using RTP-coefficients from Shindell and Faluvegi
(2010), as described in Sect. 2.2.3:

$$ARTP(ex - Arc)_{BC,r,Arc,s}(H) = \sum_{l=1}^{3} \frac{F_{l,BC,r,s}}{E_{BC,r,s}} \times RCS_{BC,l,Arc} \times R_T(H) \tag{6}$$

For the RF within the Arctic the response ($ARTP(Arc)_{BC,r,Arc,s}$) is calculated according to Eq. (7) following the
method presented in Lund et al. (2014):

$$ARTP(Arc)_{BC,r,Arc,s}(H) = \sum_{z} \frac{F(z)_{Arc,BC,r,s}}{E_{BC,r,s}} \times RCS(z)_{BC,Arc,Arc} \times R_T(H) \tag{7}$$

Both the RF ($F(z)_{Arc,BC,r,s}$) and the regional sensitivity matrix ($RCS(z)_{BC,Arc,Arc}$) have a dependence on the height of
the BC which is denoted by the z in Eq. (7). We apply a vertically-resolved regional sensitivity matrix based on
Fig. 2(a) in Lund et al. (2014), which shows the sensitivity of the Arctic surface temperature response to the altitude
of RF in the Arctic from Flanner (2013) interpolated to the vertical structure in OsloCTM2. This relationship can
be combined with the normalized BC RF from Samset and Myhre (2011) to give a normalized Arctic surface
temperature response to BC perturbations at different altitudes.
We apply the vertical profile of BC concentration in the Arctic for all three models used. These vertical profiles
are converted into RF profiles based on the vertically resolved RF to burden ratio in OsloCTM2.
Our second refinement is on the forcing-response coefficients for BC on snow in the Arctic, where we use the
forcing-response sensitivity found by Flanner (2013).
As the semi-direct effect is implicitly included in the estimates from Flanner (2013), we cannot distinguish between
direct RF and semi-direct RF for RF occurring in the Arctic. The Arctic RF due to the semi-direct effect provided



in Bellouin et al. (2016) is left out to avoid double counting. However, our argument is that the explicit vertically
resolved forcing-response relationships is a much better fit than a vertically averaged forcing-response
relationships, which makes this the preferable method. As a result, this study's ARTP estimates of the semi-direct
effect in the Arctic is due to the semi-direct RF from outside the Arctic.
The Flanner (2013) study is based on an equilibrium climate sensitivity of 0.91 K/(W m$^{-2}$), which is 14% lower
than applied in our study. We adjust our calculations so that the climate sensitivity is in line with the rest of our
calculations (Boucher and Reddy, 2008). The correction is done with a two-layer box-diffusion model based on
the parameters of the Hadley Centre model (see Aamaas et al., 2013), which also modifies the timescales of the
impulse response function.
The total response in the Arctic is then the sum of the contributions from BC forcing outside of the Arctic and
inside of the Arctic.
$$ARTP_{BC,r,Arc,s}(H) = ARTP(ex - Arc)_{BC,r,Arc,s}(H) + ARTP(Arc)_{BC,r,Arc,s}(H) \qquad (8)$$

## 3    Results

### 3.1    ARTP values

#### 3.1.1    Best estimates
**Results for ARTP(20)**

The best estimates of ARTP values for a time horizon of 20 years are presented in Fig. 1, for each of the four
emission regions, the four response bands, plus the global mean, for all emitted species considered here. We
provide values for other time horizons (10, 50 and 100 years) in Supporting Information Sect. 2. The rationale for
highlighting 20 years is that if the focus is to be placed on mitigation of SLCFs then it is more appropriate to
investigate climate impacts on short timescales. Continuous time horizons between 1-50 years are given in Sect.

235    3.1.5.

The uncertainties in Fig. 1 are given as a range following the differences in RFs estimated between the models.
We acknowledge other uncertainties, such as for climate sensitivity, which are discussed in Sect. 3.4. The
uncertainty is often larger than the variation between different emission regions, seasons, and responses in the
latitude bands. However, we will show in Sect. 3.1.4 that the relative variations between the best estimates for
individual species are often robust. As ARTP values for the shipping sector are based on only two RF estimates,
uncertainty ranges are not given for shipping. The robustness in the best estimate for shipping is likewise lower
than for the other regions. E.g., these two models disagree for shipping on the sign for the *aerosol effect* of NO$_x$
emissions. NH$_3$ estimates are also from one model only, and are not shown for shipping (because emissions from
that sector are negligible).

**Response patterns**

For emissions from a given region, the latitudinal response pattern is partly governed by the pattern of RF and
partly the pattern in the forcing-response coefficients. The RF signal is mainly located in the latitude bands near
the emission sources for the short-lived components, while it is more evenly distributed for processes linked to



methane. Hence,  as shown in Bellouin et al. (2016) (see especially their Fig. 7), emissions in Europe and East
Asia give largest RF in the NH mid-latitude band and the smallest in the Southern Hemisphere (SH) mid-high
latitudes. Due to heat transport between the latitude bands and the temperature response lasting over several years,
the forcing-response is averaged out over several latitude bands by the temperature response. Nevertheless, the
temperature response has higher sensitivity towards the Arctic and NH mid-latitude bands (see all panels in Fig.
1) as a result of local feedback processes being stronger in the Arctic, driven by local cloud, water vapor, and
surface albedo feedbacks (Boer and Yu, 2003).
We next consider differences between the emission regions Europe and East Asia. The RF per unit emission is
dependent on where the emissions occur, which causes differences in the ARTP(20) values. The differences in the
global average of RFs and global emission metric values such as AGTP(20) are discussed in Aamaas et al. (2016).
In short, the emission metric values for the aerosols are larger for European than East Asian emissions, but not for
$NH_3$ in winter. Variations are also seen for the ozone precursors, but these differences are relatively smaller
between European than East Asian emissions for CO and VOC than for the aerosols. For CO, East Asia has
marginally larger values (see Figs. 1(K) and 1(L)) and marginally larger for European VOC emissions (see Figs.
1(M) and 1(N)). The main difference in the global average of ARTP values calculated here and the AGTP values
calculated in Aamaas et al. (2016) is the much larger impact for BC deposition on snow for ARTP (see Fig. 1(B)),
as the AGTP study did not account for the increased efficacy of BC deposition on snow.
The timing of emissions also influences the RF per unit emissions. The emission metric values for the aerosol
emissions in Europe and East Asia (see Figs. 1(A)-1(F)) are larger for summer than winter, except for BC. For the
aerosols, the aerosol RF is driven by seasonal variations in the incoming solar radiation. More sunlight in local
summer results in stronger RFs (Bellouin et al., 2016). Seasonal differences in atmospheric lifetimes due to
seasonality in precipitation may also contribute. BC is discussed in detail in Sect. 3.1.2.
For the ozone precursors (see Figs. 1(I)-1(N)), the largest values occur in winter for CO (Figure 1(L)) and in
summer for VOC (Figure 1(M)). CO has a longer lifetime during local winter leading to a larger fraction of the
CO emitted being transported from the higher latitudes to the Tropics. Here, the effects of CO-oxidation on tropical
OH have the largest impacts on the methane lifetime.
The latitudinal response patterns are similar for the different species. For all the species, the response bands with
the largest ARTP values are for the responses in the NH mid-latitudes (60% of the cases) and Arctic and the band
with the least response the SH mid-high latitudes (see all panels in Fig. 1). This skewness is partly due to the
emissions occurring mainly in the NH, but the same pattern is seen for $CH_4$ (Figure 1(O)), for which the emission
location is less important. Further, the high ARTP values for the Arctic are also due to stronger local feedback
processes, leading to larger forcing-response sensitivities, while high ARTP values for the NH mid-latitudes are a
combination of high RF values per unit emission and relatively large regional climate sensitivities. The low ARTP
values for SH mid-high latitudes is caused by a combination of most emissions occurring in NH and weaker
forcing-response coefficients in SH. Let us consider OC emissions in East Asia during summer as an example with
RF mostly in one band. The RF   (see Bellouin et al., 2016) in the NH mid-latitude band is 260% above the global
average, practically zero in the SH mid-high latitude band and about 50% below the global average in the other
two bands. This skewedness is also modeled in the ARTP (see Fig. 1(C)), but with more emphasis on the Arctic.





The ARTP value for the responses in the Arctic and NH mid-latitudes is about 70% and 90% above the global
average, respectively.  In the SH mid-high latitudes response band, the ARTP value is about 20% of the global
average. At the other end of the range, emissions of $CH_4$ have a global impact due to the atmospheric lifetime of
$CH_4$ (9.7 years). The RF in the Arctic band is 35% below the global average, while 25% above in the Tropics. But
the weighing is almost opposite for the ARTP, as the Arctic response band has a ARTP value 34% above the global
average and the Tropics 13% above the average (see Fig. 1(O)). For the SH mid-high latitude response band, both
the RF and ARTP are lower than the global average, by -35% and -49%, respectively.
For most of the aerosol emissions (see Figs. 1(A)-1(F)), the ARTP values for the *aerosol effects* component are
larger for emissions in NH summer than winter, even in the Tropics for emission from both Europe and East Asia.
The only exception is $NH_3$ (Figures 1(G) and 1(H)), which has a larger ARTP value for winter than summer for
East Asian and global emissions. Longer sunlight duration in the summer hemisphere yields stronger RFs (Bellouin
et al., 2016), which impact the ARTP value for the response even in the Tropics. This general observation does
not hold for BC when we include the process "BC deposition on snow", as this process is largest in NH winter.
The ARTP(20) values shift sign for some of the latitude response bands. VOC emissions generally lead to a
warming, however, our best estimate indicates a small cooling in SH mid-high latitudes for European and East
Asian winter emissions (Figure 1(N)). The negative RF for the *aerosol effect* in this response band is driving this
cooling as the other perturbations have a small impact on the response in the SH mid-high latitudes. For the ozone
precursors, the *aerosol effects*, and the short-lived ozone effect to a smaller degree, also shift between warming
and cooling depending on the latitude response band.
**3.1.2    Variation of BC response with emission season and region**
The largest differences in ARTP(20) values are seen for BC, such as the timing of emissions (comparing Figs. 1(A)
and 1(B)) and the location during winter (comparing the different emission regions in Fig. 1(B)).
The total emission metric values of BC emissions depend on which processes are included. The direct *aerosol*
*effect* is larger for summer than winter emissions. The direct temperature response is similar for emissions
occurring in Europe, East Asia, and globally. Similarly, the semi-direct effect is most pronounced in summer as
this effect is driven by absorption of shortwave radiation. When the influence from the BC deposition on snow is
included, the ARTP value increases significantly for emissions during NH winter. For emissions in Europe, the
global temperature response to the semi-direct effect is -46% and -12% of the *aerosol effect* in summer and winter,
respectively, and the deposition on snow effect 12% and 230% of the *aerosol effect* in summer and winter,
respectively. The relative share of the deposition on snow effect is 60 % lower for winter East Asian emissions
than for winter European emissions. The semi-direct effect has a relative weight of -56% compared to the *aerosol*
*effect* for the global ARTP(20) East Asian emissions in summer and close to zero in winter. The impact of BC
deposition of snow is largest when large snow and ice covered surface areas and solar radiation at the BC
deposition location is combined, such as in late winter. The response from European emissions is larger than for
East Asian emissions since the emission region is closer to the Arctic, which makes BC transport into the sensitive
Arctic more likely (Sand et al., 2013). The effect of the BC deposition on snow dominates the winter-summer
difference for BC and hence our results are sensitive to both the calculated RF and efficacy for this BC process.





The Arctic response amplification, i.e., how much stronger the response is in the Arctic relative to the global
average, is largest for winter emissions as the deposition on snow effect is relatively larger than for summer
emissions. The total Arctic response amplification for BC is for European emissions 240% and 390% larger than
the global average in summer and winter, respectively, and for East Asian emissions 160% and 240% larger than
the global average in summer and winter, respectively. As a result, wintertime BC emissions have the largest
latitudinal variation in the ARTP(20) for all SLCFs. This Arctic amplification is driven by the temperature response
from deposition on snow effect (almost 500% for European emissions and 400% for East Asian emissions for this
process), which is largest in the Arctic response band, above the global average in the NH mid-latitude, and below
average in the two other response bands. Latitudinal response variations are also found for the other processes, but
relatively much smaller.

### 3.1.3 Comparison with Collins et al. (2013)

Our findings are largely consistent with those by Collins et al. (2013). Similarities occur because the two studies
share some of the same forcing-response coefficients and climate sensitivity (Boucher and Reddy, 2008). In this
work, we have a more detailed treatment of BC in the Arctic and we include $NH_3$, as well as more detailed for
aerosol impacts on ozone precursors. ARTP values are also given for two seasons, for the shipping sector and our
global estimate include all emissions.
The ARTP(20) values in Collins et al. (2013) are mostly lower than the average response of annual emissions in
this study, while the variations between the latitude response bands are mostly similar. We model 180% and 80%
stronger global temperature sensitivity from European and East Asian emissions of BC. The largest difference is
that our study included the response from BC deposition in snow. In addition, Collins et al. (2013) applied a
forcing-response coefficient for the BC direct RF that gives an Arctic cooling due to emissions in the Arctic
(Shindell and Faluvegi, 2009). When including a more detailed parameterization for atmospheric BC in the Arctic
that considers the height of the BC (see Sect. 2.2.4), the global temperature response of BC emissions increases
by 4-14%. The difference is much larger in the Arctic, and the increase in the Arctic is 22-210% when only
considering the BC direct and 1$^{st}$ indirect effects.

### 3.1.4 Robustness for individual species

The differences between ARTP(20) values for different emission regions and emission seasons, as well as for the
response in different latitude bands for one set of emissions, are smaller than the inter-model uncertainty ranges.
However, the ARTPs based on RFs for the individual models agree often with the best estimate on the ranking
between the different emission and response cases, which strengthens our confidence that the variations calculated
for the best estimate are robust. In Supporting Information Sect. 3, we quantify this robustness and find a high
robustness consistent with similar analysis done on AGTP(20) values (Aamaas et al., 2016). As the temperature
response is more smeared out globally for the ozone precursors than for the aerosols, the models agree to a larger
extent for the aerosols concerning which latitude response bands see the largest and smallest temperature
perturbations. For BC, we compare results only including the *aerosol effects* as only one model includes BC on
snow and semi-direct effects. The model NorESM has the largest discrepancy relative to the best estimate for $NO_x$
and VOC, while HadGEM3 disagree the most for CO.




### 3.1.5  Variations with time horizon

We have so far only analyzed ARTP(20) values. Here we present results for a range of time horizons up to 50 years in Fig. 2. The ARTP values vary greatly with time horizon and generally decrease in magnitude with time for SLCFs, especially for the aerosols (see Figs. 2(A) and 2(B) for BC). The ranking between different regions, seasons and latitude bands also changes with varying time horizon for the ozone precursors (see Figs. 2(C)-2(H)). The reason is that the aerosols and aerosol precursors have atmospheric lifetimes of about a week, while methane has an atmospheric perturbation lifetime of almost 10 years, which will lead to variations in the relative weight of the short-term and long-term processes with varying time horizons for the ozone precursors (e.g., Collins et al., 2013).

The temporal variability shows that $NO_x$ emissions in Europe have the most negative ARTP values for summer than for winter for all time horizons, which is due to a stronger methane effect (Figure 2(C)). For East Asian emissions, the situation is mixed with the most negative ARTP values in the first 10-15 years for winter emissions, while summer emissions have the most negative values for longer time horizons (Figure 2(D)). For summer emissions, ARTP values in the first few years is pushed upwards by stronger solar insolation than in winter leading to more short-lived ozone. For the ozone precursors, the ranking on which latitude band is the most sensitive is mostly unchanged after 5 years, but can vary in the first years.

### 3.2  Regional temperature response for 2008 emissions

Given the ARTP values, we calculate the regional and global temperature responses due to real-world emissions of SLCFs. The temperature response at time $H$ in latitude band $m$ for an emission $E$ of species $i$ is

$$\Delta T_{i,r,m,s}(H) = E_{i,r,s} \times ARTP_{i,r,m,s}(H) \qquad (9)$$

We estimate the temperature response in the four latitude bands for a time horizon of 20 years given real-world emissions in 2008 from Europe, East Asia, the shipping sector, and globally (Klimont et al., In prep.). The global emissions are given in Supporting Information Table S7. Such a view on regional responses is useful as regional variations will be hidden in the global mean response (e.g., Lund et al., 2012). The emissions include seasonal variability with emissions often being largest in the NH winter season. The temperature perturbations are mainly governed by the ARTP(20) values given in Sect. 3.1.1, but also by the seasonal cycle of the emissions. The emissions in Europe and East Asia are larger in winter than summer for all species except $NH_3$, driven by larger residential heating and cooking emissions during winter conditions. BC emissions are about 70% larger, OC emissions 70-100% larger, and $SO_2$ emissions almost 20% larger in East Asia and more than 40% larger in Europe (Klimont et al., In prep.). The seasonal variability is smaller for the ozone precursors, CO with the largest range (43% more in winter).

For the global source region, ignoring the seasonality by applying annually averaged emissions and ARTP values gives similar total temperature responses as treating the seasons separately and then averaging (differences of 0-3%). However, when treating Europe or East Asia individually seasonal information changes the temperature estimates by up to 18%. The difference is largest for the aerosols. For Europe, the temperature response increases by 8% for BC and decreases the cooling by OC by 10%. The largest relative changes are seen in the net temperature perturbation of all SLCFs.




Figure 3 shows that the temperature perturbations are smallest for the SH mid-high response latitudes and largest
for the Arctic and NH mid-latitudes, as seen for ARTP(20). For most latitude response bands, $SO_2$ has the largest
impact, so the net effect of the seven SLCFs is a cooling in most of the cases. BC has the second largest impact
with a warming that is largest for winter emissions. The shipping sector is dominated by cooling from $SO_2$ and
$NO_x$ (see Figs. 3(E) and 3(F)), while the other sectors have a much broader mix of species causing both heating
and cooling. However, $NO_x$ can be both warming and cooling depending on emission metric choices. For ARTP(20)
applying sustained emissions, $NO_x$ has a relatively smaller cooling impact and even contributes to warming in
some latitude bands for shipping emissions in summer (see Supporting Information Fig. S1).
Emission of SLCFs lead normally to net cooling or effects that cancel each other out. However, we show that some
specific cases cause warming in the Arctic (see Figs. 3(B), 3(D), and 3(H)). Winter emissions in Europe and East
Asia cause a warming in the Arctic and almost no net perturbation in the NH mid-latitudes and other bands. The
main reason is the strong heating from the BC deposition on snow for winter emissions close to snow and ice
surfaces, as well as relatively larger BC emissions in winter than for the other species. For summer emissions in
Europe and East Asia (Figures 3(A) and 3(C)), the situation is the opposite with the largest cooling in the Arctic
and NH mid-latitudes. A small net heating in the Arctic is also observed for global emissions in the NH winter
season.

### 3.3 Global temperature response and comparing ARTP and AGTP

We discuss how adding complexity with four latitudinal response bands impacts the metric value by comparing
the global temperature response for regional and seasonal emissions presented in Sect. 3.2 based on ARTP with
the  AGTP calculation in Aamaas et al. (2016). Shindell (2014) concluded that the efficacy of the temperature
response depends on the location of the RF. As a result, more RF in the NH middle to high latitudes for the aerosols
give a larger response than a globally averaged RF. Lund et al. (2012) found that an emission metric first based on
regional variations, then averaged globally gives a more complete and informative value than one based on global
mean inputs. Work by Stohl et al. (2015) shows that regional temperature estimates based on ARTPs mostly agree
with calculations with earth system models. Although heterogeneity can be better included in temperature
responses given by ARTPs compared to AGTPs, the superiority of ARTPs relative to AGTPs has not been tested
thoroughly and confirmed. However, we argue that the global temperature response can be better quantified with
ARTPs than AGTPs since a simple representation of varying efficacies due to heterogeneous RF is included.
How the global temperature responses are calculated given the AGTP values is shown in Supporting Information
Sect. 6 and Aamaas et al. (2016). For the ARTP values, the global temperature is calculated from the area-weighted
mean of the responses in the latitude bands. As the ARTP calculations are based on an efficacy of 3 for BC
deposition on snow, the same efficacy is applied in the AGTP calculations. Our comparison between the methods
applying ARTP and AGTP uses a pulse emission $E$. The difference in the global temperature perturbation ($\Delta T(diff)$)
for species $i$ between the two methods is then
$$\Delta T(diff)_{i,r,s}(H) = \sum_m E_{i,r,m,s} \times ARTP_{i,r,m,s}(H) - E_{i,r,s} \times AGTP_{i,r,s}(H),\qquad(10)$$
which is applied for each emission region $r$ and emission season $s$.




434 We compare the temperature perturbation based on ARTP and AGTP for a time horizon of 20 years using the 2008
435 emissions. The largest difference is for NH summer emissions. For global NH summer emissions, ARTP(20) result
436 in 17% more net cooling than AGTP(20) and about 26% and 32% more cooling for European and East Asian
437 emissions, respectively. The differences in responses are smaller for NH winter emissions. Annually, global
438 emissions lead to a 13% larger cooling based on ARTP than on AGTP. See Sect. 7 in Supporting Information for
439 further details. The differences emerge because the patterns of RF and efficacy are correlated, with highest RFs
440 and highest efficacies in the northern mid latitudes and Arctic. Thus, the ARTPs are necessary even to obtain a
441 global temperature response since they account for these correlations.

442 Next, we analyze the differences between applying ARTP and AGTP for the individual species (see Fig. 4 and Fig.
443 S3 in Supporting Information). The relative differences are in most cases similar for the different emission regions
444 and seasons, which show that the differences between ARTP and AGTP are governed by differences in the forcing-
445 response coefficients between the two. The relative differences are generally larger for the aerosols than the ozone
446 precursors, as seen in Fig. 4, where only the emissions regions and seasons with a relative difference larger than
447 20% are presented. The temperature responses are generally stronger for the scattering aerosols and the BC
448 deposition on snow given the ARTP than the AGTPs, which is in line with greater efficacies due to rapid and
449 strong feedbacks for RFs in the northern mid-latitudes and the Arctic latitude bands (Shindell, 2014). BC and
450 ozone precursors are in general given lower weight when using ARTPs than AGTPs. Application of ARTP and
451 AGTP values give variation of up to 30% for individual processes, with an average of 12% for individual species.
452 ARTPs are more detailed in nature and through accounting for variations in efficacy will give more realistic global
453 temperature responses.

### 3.4 Uncertainties

455 The ARTP values calculated have uncertainties and limitations given by the uncertainties in each parameter on the
456 right hand side of Eq. (1). The uncertainty ranges shown in Fig. 1 are based on the range in $\frac{F_{L,i}(t)}{E_i}$ across all
457 contributing models. Bellouin et al. (2016) point out four important aspects regarding model diversity. Lifetime
458 diversity is large, the unperturbed baseline causes diversity for non-linear mechanisms, the number of species
459 included varies among the models, and finally the strength of the interactions between aerosols and chemistry
460 differs among the models. The climate sensitivity included in $R$ is 3.9 K for a doubling of $CO_2$ concentration
461 (Boucher and Reddy, 2008); however, IPCC (2013) estimate the climate sensitivity to likely be in the range 1.5-
462 4.5K. Uncertainty is also found in the time evolution of $R_T$. We have based this impulse response function on only
463 one model, while Olivié and Peters (2013) have shown that this will vary between models. For instance, they found
464 a spread in the GTP(20) value of black carbon of about −60 to +80% due to variability for $R_T$ between models.
465 However, the uncertainty in $R_T$ is less relevant for the regional patterns. The forcing-response coefficients are also
466 based mainly on one model (Shindell and Faluvegi, 2010). While we separate between emissions occurring during
467 NH summer and winter season, forcing-response coefficients do not exist on a seasonal basis. Hence, the seasonal
468 differences presented here in the ARTP values are not due to potential differences in the response sensitivities, but
469 due to differences in the RF. Aamaas et al. (2016) observed that estimates of $\frac{F_{L,i}(t)}{E_i}$ tend to be correlated, which
470 increases the uncertainty when a mitigation package is considered.





Ideally, calculations of the temperature response of changed emissions of SLCFs should use earth system models
for the most correct estimates. However, this is extremely time consuming, and many emission perturbations will
have small signal/noise ratios. Users of emission metrics, such as policymakers and decision makers, might not
have the needed expertise to utilize advanced models. Although the ARTP calculations are simplifications and
contain uncertainties, these emission metrics are useful, simple, and quick approximations for calculating the
temperature response in the different latitude bands for emissions of single species or a mix of SLCFs.
**4    Conclusion**
We have presented ARTP values in four latitude bands (90-28° S, 28° S-28° N, 28-60° N, and 60-90° N) for
several SLCFs (BC, OC, $SO_2$, $NH_3$, $NO_x$, CO, VOC, and $CH_4$) based on four different models. Numbers are
provided for emission occurring in Europe, East Asia, from the global shipping sector, as well as globally.
Emissions were separated between the NH summer and winter seasons. Although ARTPs are simplifications, they
are useful for analyzing the temperature response to possible mitigation strategies. The ARTP values are largest
in the response bands Arctic and NH mid-latitudes and the smallest in the SH mid-high latitudes. The different
models agree in most of the cases on the ranking of the temperature perturbation in the different latitude bands.
BC is the species that is the most sensitive to the timing of emissions, to the location during winter, as well as
having the largest spread in responses between the latitude response bands in winter. The relative difference
between the response bands is largest for BC emissions during NH winter, and the more the closer to the Arctic
the emissions occur. The Arctic temperature response is 390% and 240% larger than the global temperature
response for winter emissions in Europe and East Asia, respectively. BC deposition on snow is the most important
process influencing the Arctic for BC emissions occurring in NH winter, both in absolute and relative terms.
We have also investigated how the global response based on ARTP compares with AGTP. Our study indicate that
the global temperature response can be better quantified with ARTPs than AGTPs since ARTPs include a simple
representation of varying efficacies due to heterogeneous RFs. For global emissions of SLCFs excluding $CH_4$,
calculations based on ARTP values give 13% larger cooling than based on AGTP values.  Globally, both these
calculations based on ARTP(20) and AGTP(20) show a cooling, while European and East Asian winter emissions
give a small net warming or near zero impact according to ARTP. This is driven by net warming in the Arctic and
close to zero perturbation in the other latitude bands. For summer emissions, net cooling occurs in all latitude
bands, but are largest in the NH mid-latitudes and Arctic. Seasonal emissions and seasonal ARTP values give
almost the same total temperature response as annual emissions and annual ARTP values for global emissions, but
changes the temperature responses up to 18% when looking at emissions from regions such as Europe and East
Asia.
**Acknowledgements**
The authors would like to acknowledge the support from the European Union Seventh Framework Programme
(FP7/2007-2013) under grant agreement no 282688 – ECLIPSE, as well as funding by the Norwegian Research
Council within the project "the Role of Short-Lived Climate Forcers in the Global Climate Regime". We thank
Nicolas Bellouin for providing RF data for all the models. In addition, we show our appreciation to Nicolas
Bellouin, Marianne Tronstad Lund, and Dirk Olivié for giving us vertical distributions of BC in the Arctic.



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





Table 1: The models and species included. Models are either general circulation models (GCM) or chemistry transport models
(CTM). The resolution column shows the horizontal resolution and the number of vertical layers.

| Model | Type | Resolution | BC | OC | SO$_2$ | NH$_3$ | NO$_x$ | CO | VOC | CH$_4$ | References |
|---|---|---|---|---|---|---|---|---|---|---|---|
| ECHAM6-HAMMOZ | GCM | 1.8°x1.8° L31 | X | X | X | | | | | | Stevens et al. (2013) |
| HadGEM3-GLOMAP | GCM | 1.8°x1.2° L38 | X | X | X | | X | X | X | X | Hewitt et al. (2011) |
| NorESM | GCM | 1.9°x2.5° L26 | X | X | X | | X | X | X | X | Bentsen et al. (2013);Iversen et al. (2013) |
| OsloCTM2 | CTM | 2.8°x2.8° L60 | X | X | X | X | X | X | X | X | Søvde et al. (2008);Myhre et al. (2009) |















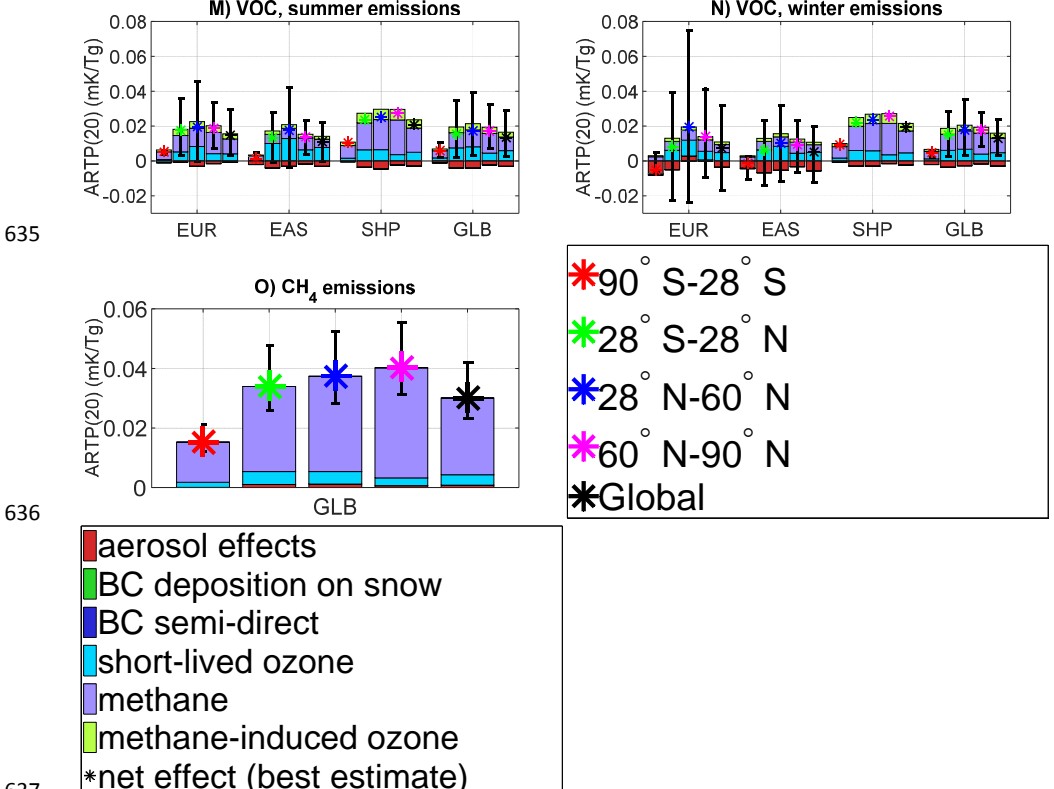




Figure 1: ARTP20 for emissions from Europe, East Asia, shipping, and global and for summer and winter. In each frame, and

for each emission region, the ARTP20 values for the four latitudinal response bands from south (left) to north (right), as well

as the global response average (rightmost), for the species, decomposed by processes. The net response is shown by the asterisk.

The regions included are Europe (EUR), East Asia (EAS), shipping (SHP), and global (GLB), all for both NH summer, May-

October (left), and NH winter, November-April (right). The uncertainty bars show the range across models, which is not given

for shipping as the best estimate is based on only two models for that sector. Due to the methodology applied, a fraction of the

semi-direct effect for BC in the Arctic is included in the *aerosol effects* process, as explained in Sect. 2.2.4. Note that the

vertical axis varies between different emitted components.





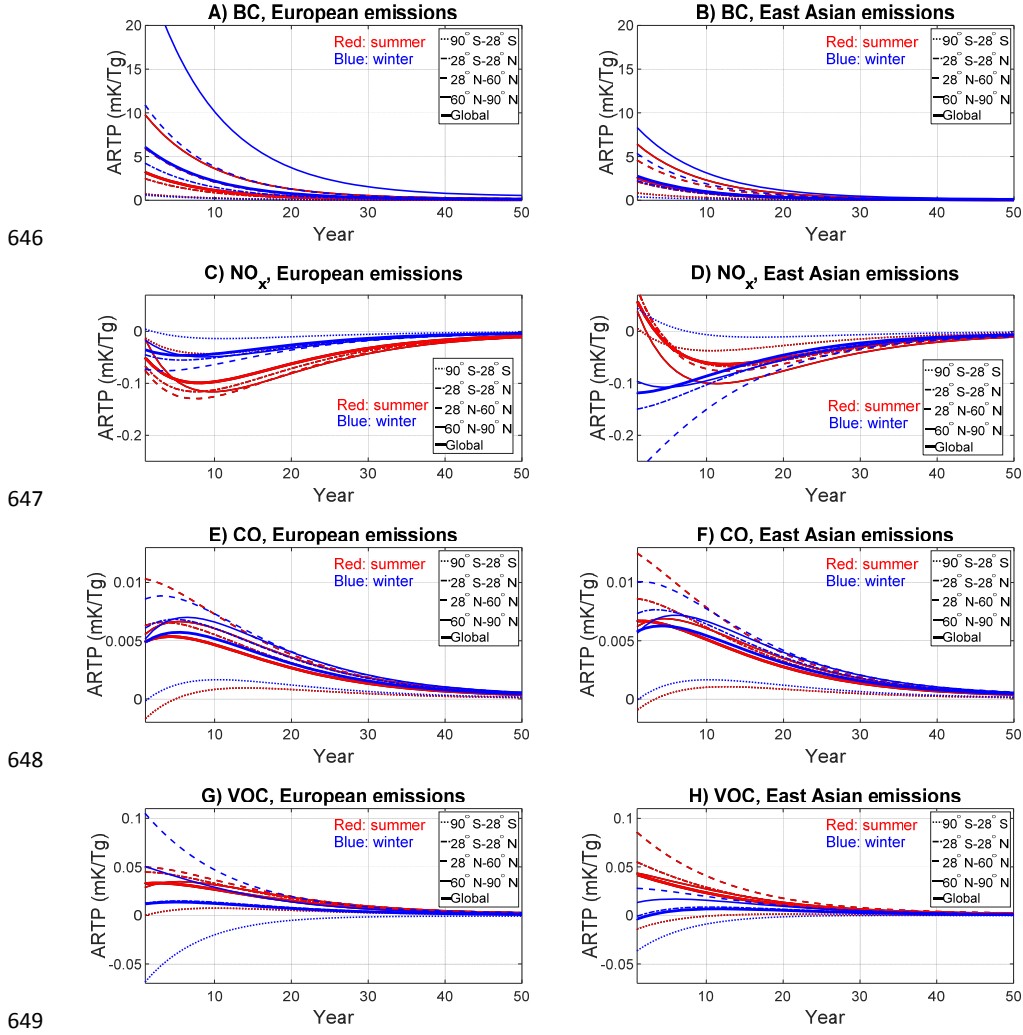

Figure 2: ARTP values in different response bands for BC and the ozone precursors for time horizons up to 50 years. Emissions
in Europe (left) and East Asia (right) in NH summer (May-October) are given as red and in NH winter (November-April) as
blue.





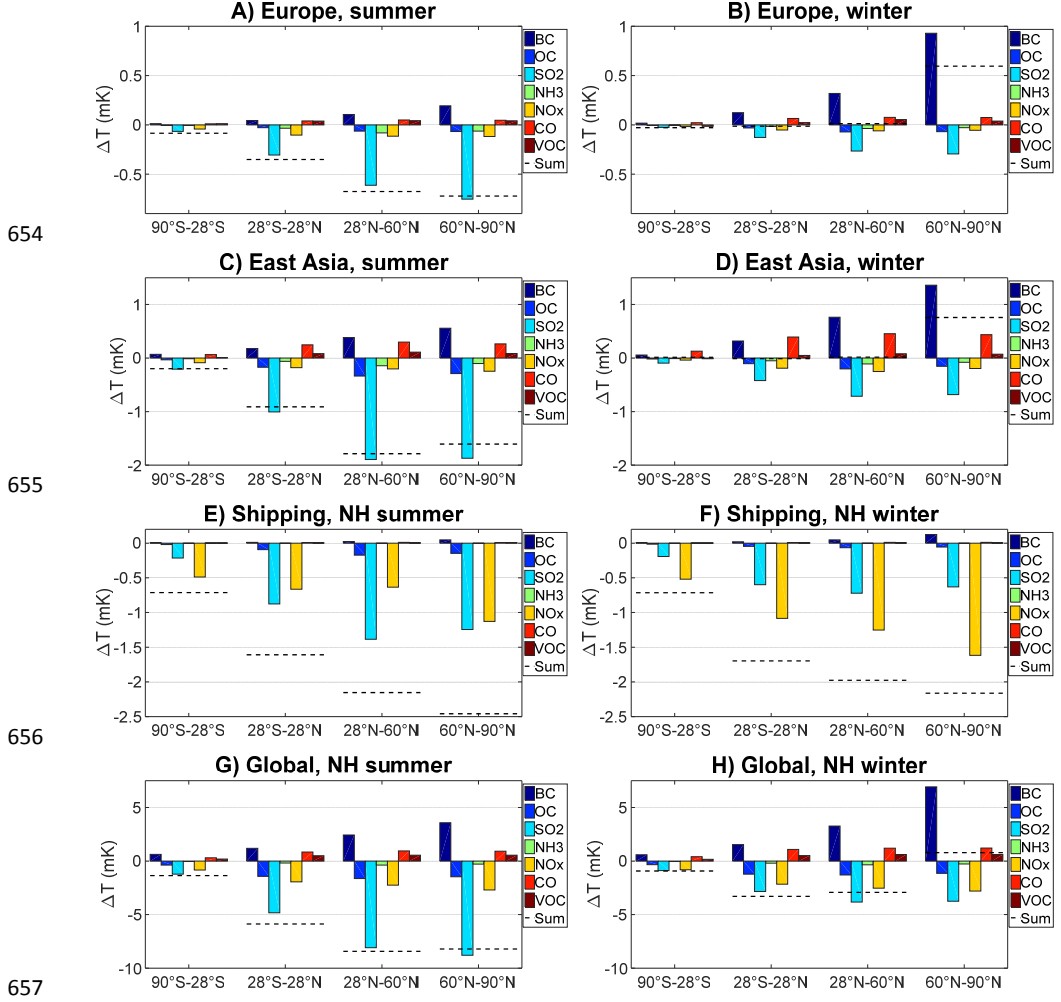

Figure 3: The regional temperature response for a time horizon of 20 years after regional and seasonal emissions in 2008 based on ARTP(20). The four latitude response bands represent the SH mid-high latitudes, Tropics, NH mid-latitudes, and Arctic. The global response average is given in Fig. S2. From top to bottom, the emission regions are Europe, East Asia, the global shipping sector, and global. The emissions are split into NH summer season (May-October) to the left and NH winter season (November-April) to the right. Note that the y-axis differs for the regions. The horizontal dashed lines show the sum for each response band.





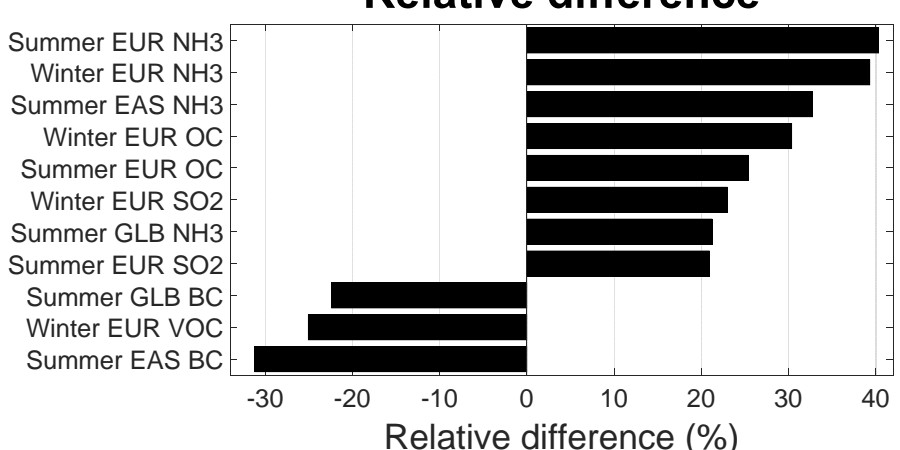

665

Figure 4: The relative difference between the global temperature responses based on ARTP and AGTP methods for a time
horizon of 20 years. Only cases with larger relative differences than 20% are shown. Positive numbers occur when the
magnitude of the global temperature response is larger when based on ARTP than on AGTP, negative when the magnitude is
largest based on AGTP.

670