# Peer review of "Regional temperature change potentials for short lived"

_Atmospheric Chemistry and Physics, 2017_

## Referee Comment (RC1) · Anonymous Referee #4 · 11 Apr 2017

General comments

The paper is interesting and in general well-written. As it is clearly described in the paper it builds on the work by Collins et al (2013), make use of a methodology developed in Shindell & Faluvegi (2009) and with data largely from Bellouin et al (2016). The study makes contributions regarding the estimation of Absolute Regional Temperature Potentials for NH3, the effect of aerosols on the ARTP for O3 precursors, extended analysis of the warming effect of BC on snow and ice and perhaps most importantly they analyze summer and winter specific metrics.

My key comment relates to the division of metrics into four latitude bands. I understand that the paper here builds on the framework by Shindell & Faluvegi (2009), but why is it 4 latitude bands, and not 6, 8 or any other number of latitude bands that is a relevant

separation for the ARTPs? This needs to be discussed and problematized. More importantly, why is not a separation between temperatures impacts on land surfaces and ocean surfaces used? The land-ocean separation may be critical for regionalized metrics due to the significant land-ocean warming contrast (Joshi et al, 2008; Boer, 2011) and the very different climate impacts on land and ocean areas. I do not think the authors need to change their calculations, but a discussion regarding the relevance of the approach they take, what important aspects they miss with the regionalization they use and how the regionalization can be developed further is needed in the paper. Finally, overall I think the paper is a valuable contribution to the scientific literature and deserves to be published after the general comments above and the specific comments below have been taken into account.

Specific and technical comments

1. Page 1 line 31-32. The authors write "CH4 is often also included because its lifetime of around 10 years is shorter than or comparable to climate response timescales." I am not sure if I have seen this argument before. Please, justify with a reference why this is the reason why CH4 is included among the short lived climate forcers.

2. Page 4 line 131-132. The authors write "We assume that the time evoluation of temperature in each response band follows the global mean temperature". Is this a valid assumption? For example, Cherubini et al (2016) do a related, but brief analysis using MAGICC, where they estimate regional metrics based on emissions that take place in either NH-ocean, NH-land, SH-ocean and SH-land, and where the NH and SH temperature response is analysed. If one contrast the assumption that the "time evaluation of temperature in each response band follows the global mean temperature" with results presented in Cherubini et al (2016) figure 3 the assumptions appears to be rather crude, especially on short time scales. The authors need in greater length justify this assumption.

3. Equation 2. In equation 2 the indices r,m,s, for the ARTP is dropped. Why? Please,

be consistent throughout the equations or explain carefully difference between similar variables used in different equations.

4. Equation 3 and line 14-143. The authors write: "the general expression for the ARTP can be simplified to". Even though the mathematics behind this approximation is quite simple it should be shown and/or explained in a footnote, in the supplementary material or a reference to a paper where this is done should be included.

5. Page 5 line 152. The authors mention that the average adjustment time of CH4 is 9.7 years in the three models used. This is relatively short compared to the IPCC AR5 assumption (12.4 years). Can the authors explain why a relatively short atmospheric adjustment time is find in the models used in the paper?

6. Page 5 lines 164-165. The authors write "RCS matrices only exist for annual emissions, we assume we can apply the same set of matrices for 165 emissions during NH summer and winter." Please justify this assumption.

7. Page 8 lines 277-279. The authors write "For all the species, the response bands with the largest ARTP values are for the responses in the NH mid-latitudes (60% of the cases) and Arctic and the band with the least response the SH mid-high latitudes (see all panels in Fig. 1). This skewness is partly due to the emissions occurring mainly in the NH, but the same pattern is seen for CH4 (Figure 1(O)), for which the emission location is less important." The argument "This skewness is partly due to the emissions occurring mainly in the NH" is confusing. I first thought that the authors were referring to actual real world emissions, but that is totally irrelevant since you study equally sized emission pulses from different regions. Please write clearly what you mean with "This skewness is partly due to the emissions occurring mainly in the NH". There is also a similar argument in line 282 where the authors write "most emissions occurring in NH". Please clarify!

8. Figure 4 and page 13 line 445-447. The authors write "The relative differences are generally larger for the aerosols than the ozone precursors, as seen in Fig. 4, where

only the emissions regions and seasons with a relative difference larger than 20% are presented." Why is only cases where the relative difference between ARTP and AGTP are larger than 20% shown? Wouldn't it be equally relevant to see the cases where the difference is small?

References

Joshi, M.M., Gregory, J.M., Webb, M.J. et al., (2008) Mechanisms for the land/sea warming contrast exhibited by simulations of climate change. Clim Dyn 30: 455. doi:10.1007/s00382-007-0306-1

Boer, G.J. (2011) The ratio of land to ocean temperature change under global warming. Clim Dyn 37: 2253. doi:10.1007/s00382-011-1112-3

Cherubini F., J. Fuglestvedt, T. Gasser, A. Reisinger, O. Cavalett, M.A.J. Huijbregts, D.J.A. Johansson, S.V. Jørgensen, M. Raugei, G. Schivley, A. Hammer Strømman, K. Tanaka, A. Levasseur (2016) Bridging the gap between impact assessment methods and climate science. Environmental Science & Policy 64: 129-140

For other references please see the paper by Aamaas et al, 2017

---

## Referee Comment (RC2) · Anonymous Referee #1 · 16 Apr 2017

The paper presents estimates of regional temperature change in latitude bands from regional emissions of several different short-lived climate forcers based on radiative forcing calculations and regional climate sensitivities from litterature. The paper is well presented and the results are useful for first order assessment of the climate impacts of different mitigation strategies for air pollutants. The paper merits publication in ACP after considering the following comments.

Section 3.4 summarizes different uncertainties. It would be useful if the authors could extend the section to include also a discussion of what research that is most urgently needed to reduce the uncertainties in the methodology used.

p9 l319 replace of with on

p10 l337 suggest "more detailed estimates for"

p11 l370-371 check language

---

## Referee Comment (RC3) · Anonymous Referee #2 · 23 Apr 2017

The manuscript by Aamaas et al. presents new calculations of regional temperature change potentials (ARTPs) calculated using radiative forcing estimates from four different models, and regional (actually zonal) climate sensitivities from one model, all taken from past studies. In a way it replicates the work of Collins et al. (2013), though with different data used for the radiative forcing estimates. It also includes some methodological advances compared to that previous work, such as the separation of the impact of different seasons, and accounting for the vertical structure of BC. The paper is certainly within the scope of ACP. It does not include any major specific new findings (those have been documented in earlier papers on which it is based), but it will be a useful addition to the literature when it comes to exploring the development and application of regional emission metrics. Therefore, I suggest its publication following the revisions and clarifications suggested below.

[Figure]

GENERAL COMMENTS:

- I find the title somewhat misleading. It suggests that multiple models were used, but for the actual temperature response, the calculations still rely on one model. I suggest that this kind of title is kept for when the community has RCSs from more than one model, and therefore I recommend removing that part of the title of the current manuscript.

- I suppose if a policy maker (or a scientist) was in need of regional metrics, they might end up being confused as to whether they should use those presented here or those presented in Collins et al. (2013). The models used for radiative forcing estimates here may be somewhat newer, but they are also fewer. Is there anything convincing that could be said (perhaps in Sect. 3.1.3) as to which of the estimates is more reliable, specifically when it comes to the radiative forcing terms? Also, it would have been interesting to see how the numbers in this study would have differed had the same method as Collins et al. (2013) been used here (i.e. without the methodological advances), but I appreciate that this may be a quite substantial task at this stage.

SPECIFIC COMMENTS:

Page 1, Line 14: Suggest changing "the globe" to "the entire globe".

Page 1, Lines 21-22: Sentence not entirely clear.

Page 1, Line 31: Suggest rephrasing to "included in the definition because".

Page 1, Line 32: The temporal variation has changed over time?

Page 2, Line 43: component -> constituent.

Page 2, Line 54: I do not think that all the papers referenced here (e.g. Stevenson et al. (2005), Wild et al. (2001), Fry et al. (2012)) quantify the global temperature response to emissions broken down by region. But that is what the reader is left to think.

Page 2, Lines 55-57: Strictly speaking, Shindell and Faluvegi (2010) did not present

regional temperature potentials; a potential is a response per unit emissions, whereas their paper provided responses per unit forcing.

Page 2, Lines 67-68: I think you need to explain up front to the reader what the differences are with Aamaas et al. (2016).

Page 3, Line 73: Applied to do what?

Page 3, Line 82: Perhaps rephrase to "The regional RFs are then averaged for four latitude bands..."?

Page 3, Line 86: Perhaps "effects" is a slightly better choice of word here than "processes". Also, I would say it is worth briefly mentioning/listing those effects here in the sentence so that the reader does not have to necessarily look at the figure to see what is meant.

Page 3, Lines 89-90: So, these are instantaneous forcings?

Page 3, Line 94: "methane induced" -> "methane-induced".

Page 3, Lines 100-102: Not clear how such an experiment can diagnose the semi-direct effect alone. Imposing a perturbed concentration of BC would have both direct and semi-direct effects, no?

Page 3, Lines 102-104: Not clear what ozone and methane have to do with aerosol direct and 1st indirect effects.

Page 3, Line 108: So what is ECHAM6 used for in this study?

Sect. 2.2: What are the implications of using RCSs derived from equilibrium simulations to infer metrics for transient situations?

Page 4, Line 129: "Pattern" implies "geographical distribution", so I suggest replacing with e.g. "ratios".

Page 3, Line 138: It is not mentioned what t' represents. Presumably it is the timing

of emission. Also, the indexing of location and season is suddenly dropped here. It should be mentioned in the sentence before the equation that this equation holds for the impact of every region/season or summations need to be added around the integral.

Page 5, Line 154: Shouldn't the upper limit of integration be H?

Page 5, Line 160: do -> apply.

Page 5, Lines 162-163: What is meant here by "are only incorporated in RT"?

Page 5, Line 166: What does CO2 have to do with scattering aerosols?

Page 5, Line 167: I suggest changing "long-lived" to "longer lived", as, from a climate perspective, "long-lived" implies something even longer.

Page 5, Lines 165-169: I am not at all sure what has been done here. Shindell and Faluvegi (2009) provide values for every individual effect, i.e. sulphate (proxy for all scattering aerosols), ozone, BC, CO2, methane. So why haven't those simply been used here?

Page 5, Line 170: "based on several sources" is vague here.

Page 5, Line 171: Is it really for the "aerosol effects", or for the BC part of the aerosol effects?

Page 5, Line 173: Again, what does CO2 have to do? I may be missing something, but I guess so will several other readers, since this is not explained clearly.

Page 6, Line 187: do -> apply.

Page 6, Lines 190-193: A few references are needed to support these statements.

Page 6, Line 212: This is somewhat confusing. Since Flanner (2013) is used for the estimates of sensitivities for the case of BC on snow effects, how can the semi-direct effect be implicitly included?

Page 7, Line 234: Suggest rephrasing to "Results for continuous time horizons...".

Page 7, Line 242: regions -> changes.

Page 8, Line 252: Not clear what is meant.

Page 8, Line 265: Is the increased efficacy a consequence of accounting for the vertical structure? If so, worth mentioning.

Page 8, Lines 273-274: (Last sentence in paragraph) On this timescale only, right? Also, this seems to also hold for NOx; perhaps worth mentioning?

Page 8, Lines 277-278: Emissions should not matter since the metrics are normalised by emissions, right?

Page 8, Lines 280-281: OK, but why? Shindell et al. (2015) provide some insight worth discussing.

Page 9, Line 299: Due to the largest presence of snow in the winter in the NH, presumably?

Page 9, Line 302: Why is there this negative aerosol response to VOCs?

Page 9, Line 308: Is it really the emissions region that drives larger differences than for other species, or the response region? It seems that the latter is the case.

Page 9, Line 319: of -> on

Page 10, Line 329: for -> among

Page 10, Line 336: Shouldn't Shindell and Faluvegi (2009) be cited here?

Page 10, Lines 343-345: Whereas what is the case here?

Page 10, Line 360: disagree -> disagrees

Page 11, Lines 370-371: Is this because in the summer there is more OH expected per unit NOx change, due to higher insolation?

Page 11, Line 380: Link with the earlier Equation 2.

Page 11, Line 382: Pulse emissions, not sustained, right?

Page 11, Lines 388-390: Larger than what? Presumably it refers to winter vs summer, but it needs to be clarified.

Page 11, Lines 390-391: All ozone precursors or just CO?

Page 12, Line 406: A bit confusing that earlier methane was included and here it is not, especially since often the "short-lived" terminology includes methane. Anyway, apparently methane has been dropped in this section, and it has to be mentioned upfront in it that it is not accounted for.

Page 12, Line 410: Larger even than those of sulphate? I doubt it. Probably it is meant that there is a larger seasonality for BC than for other species.

Page 13, Line 444: Perhaps add "for the same species" after "seasons".

Page 13, Line 456: Uncertainty in emissions is not accounted for, right? In which case $E_i$ should be removed from the fraction shown in this sentence.

Page 13, Line 469: Correlated with what?

Page 14, Line 476: And not just for SLCFs, right? Given that WMGHGs also cause regionally varying effects.

Sect. 3.4 (general): And what about the propagation of uncertainties in RCS due to internal variability, as reported in Shindell and Faluvegi (2009)? And also uncertainties due to spatial variability and subsequent averaging? Not that it would be expected to account for them at this stage, but worth mentioning and perhaps speculating on their importance.

Page 14, Line 487: Suggest removing "the more".

Page 14, Line 491: indicate -> indicates

Page 14, Line 500: Suggest adding "by" before "up" and "individual" before "regions".

REFERENCES:

Shindell, D. T., G. Faluvegi, L. Rotstayn, and G. Milly (2015), Spatial patterns of radiative forcing and surface temperature response, J. Geophys. Res. Atmos., 120, 5385–5403, doi:10.1002/2014JD022752.

---

## Short Comment (SC1) · 9 May 2017

The manuscript by Aamaas et al. (2017) evaluates the impact of emissions changes resolved across four latitudinal bands on temperature change. Overall, this is a very valuable work, and we appreciate several aspects of this paper, especially the multi-model efforts and exploration of the role of altitude for BC. However, discussion of the works cited in this comment and associated caveats on the use of regional temperature potentials should be included.

The use of latitudinal bands stems from the work of Shindell and Faluvegi (2009) and Shindell (2012), which evaluated climate response to radiative forcing in these bands. However, we take issue with the application of these bands for defining relationships between emissions and radiative forcing, which ignores previously published

work quantifying variability of radiative forcing efficiencies within these bands of more than an order of magnitude.

As noted by another reviewer, these regions are presented in Aamaas et al. (2017) arbitrarily in the context of relating emissions to radiative forcing, without consideration of more highly resolved regions, or – even more importantly – resolving emissions at scales other than latitudinal bands.

Further, we have in our own work explicitly shown that radiative forcing efficiencies of aerosol (Henze et al., 2012; Lacey et al., 2015) and ozone (Bowman and Henze, 2012) precursor emissions vary tremendously – more than 1000% – across latitudes. For aerosols, the key features modulating radiative forcing efficiency are related to aerosol lifetime over surfaces of varying albedo and the chemical environment for forming secondary PM from gas-phase precursors (such as the ratio of ammonia to sulfate and nitric acid). Latitude has little bearing on aerosol radiative forcing efficiency, although (the Himalayan region aside) it does impact the indirect effects of BC deposition on snow and ice (Lacey et al., 2015). For short term ozone direct radiative forcing (DRF) efficiency, latitude is a key variable, but also factors such as atmospheric chemistry, altitude, and the efficiency of vertical mixing play important roles. For example, Bowman and Henze (2012) find that "NOx emissions in Chicago would lead to 0.01 mW/m$^2$ change in DRF but the equivalent absolute reduction to emissions east of Atlanta would lead to a 0.035 mW/m$^2$ DRF reduction."

We hope that a revised manuscript from Aamass et al. will consider these important factors in their presentation and evaluation of regional temperature potentials, explicitly stating the uncertainties in application of their coefficients to evaluate temperature impacts of emissions changes at scales other than latitudinal bins. For example, while the latitudinal dependency of climate sensitivities imparts a strong-latitudinal dependence on the overall regional temperature potentials for emissions, the sub-latitudinal impact of emissions on radiative forcing can lead to important differences in the temperature impacts of equivalent changes to emissions in country-scale cookstove mitigation scenarios (Lacey and Henze, 2015; Lacey et al., 2017).

Daven K. Henze, Associate Professor, Dept of Mechanical Engineering, University of Colorado at Boulder. Boulder, CO, USA.

Forrest Lacey, Ph.D., Postdoctoral Researcher. National Center for Atmospheric Research (NCAR), Boulder, CO, USA.

Bowman, K. W., and D. K. Henze (2012), Attribution of direct ozone radiative forcing to spatially-resolved emissions, Geophys. Res. Lett., 39, L22704, doi:10.1029/2012GL053274.

Henze, D. K., D. T. Shindell, F. Akhtar, R. J. D. Spurr, R. W. Pinder, D. Loughlin, M. Kopacz, K. Singh, and C. Shim (2012), Spatially refined aerosol direct radiative forcing efficiencies, Environ. Sci. Technol., 46, 9511 - 9518, dx.doi.org/10.1021/es301993s.

Lacey, F., and D. K. Henze (2015), Global climate impacts of country-level primary carbonaceous aerosol from solid-fuel cookstove emissions, Environ. Res. Lett., 10, 114003, doi:10.1088/1748-9326/10/11/114003.

Lacey, F., D. K. Henze, C. Lee, A. van Donkelaar, R. V. Martin (2017), Transient climate and ambient health impacts due to national solid fuel cookstove emissions, Proc. Nat. Acad. Soc., doi:10.1073/pnas.1612430114.

———————————————————————

---

## Author Comment (AC1) · 21 Jun 2017

We thank all the reviewers for helpful comments that will improve our manuscript. Our responses are given in red.

**Anonymous Referee #4**

General comments

The paper is interesting and in general well-written. As it is clearly described in the paper it builds on the work by Collins et al (2013), make use of a methodology developed in Shindell & Faluvegi (2009) and with data largely from Bellouin et al (2016). The study makes contributions regarding the estimation of Absolute Regional Temperature Potentials for NH3, the effect of aerosols on the ARTP for O3 precursors, extended analysis of the warming effect of BC on snow and ice and perhaps most importantly they analyze summer and winter specific metrics.

My key comment relates to the division of metrics into four latitude bands. I understand that the paper here builds on the framework by Shindell & Faluvegi (2009), but why is it 4 latitude bands, and not 6, 8 or any other number of latitude bands that is a relevant separation for the ARTPs? This needs to be discussed and problematized. More importantly, why is not a separation between temperatures impacts on land surfaces and ocean surfaces used? The land-ocean separation may be critical for regionalized metrics due to the significant land-ocean warming contrast (Joshi et al, 2008; Boer, 2011) and the very different climate impacts on land and ocean areas. I do not think the authors need to change their calculations, but a discussion regarding the relevance of the approach they take, what important aspects they miss with the regionalization they use and how the regionalization can be developed further is needed in the paper. Finally, overall I think the paper is a valuable contribution to the scientific literature and deserves to be published after the general comments above and the specific comments below have been taken into account.

The first order answer on why we use four latitude bands is that we base our calculations on the literature. The reviewer is right that it would be preferable to do our calculations on a more detailed level, but such a framework does not exist at the time. In response to reviewer 1, we have added one paragraph in Section 3.4 about what research we would like. Research on that would be highly welcome, but out of the scope for our study. Some of the reason why Shindell & Faluvegi (2009) did four latitude bands is the relative mixing time in the meridional direction versus the zonal direction. Shindell et al. (2010b) find that responses to inhomogeneous forcing extends roughly 3500 km or 30° in the meridional direction versus more than 10 000 km in the zonal direction. We have added to Section 2.2:

"Our study separates between four latitude response bands, in line with the typical width of response bands to inhomogeneous forcing found by Shindell et al. (2010), while more detailed modelling will be possible with a finer-masked RCS matrix available."

We have added a paragraph in the uncertainty section (3.4) on the land-ocean issue. This could potentially be done in future research, but should probably consider differences between the different species.

"The temperature response will vary by location, such as land surface versus ocean surface. These differences are not accounted for in our study, but the increased efficacy in the RCS matrix towards the

NH can be partly attributed to larger land area fraction in the NH (Shindell et al., 2015). The temperature increase is in general larger over land than ocean (Boer, 2011) driven by several local feedbacks (Joshi et al., 2008). We do not have data to break down this effect for our emission regions, but results in Shindell (2012) indicate that the land response may be 20 % larger than the average."

We have also added a section on what research is most needed to reduce uncertainty, as a response to reviewer 1.

Shindell, D., Schulz, M., Ming, Y., Takemura, T., Faluvegi, G., and Ramaswamy, V.: Spatial scales of climate response to inhomogeneous radiative forcing, Journal of Geophysical Research: Atmospheres, 115, D19110, 10.1029/2010JD014108, 2010.

Specific and technical comments

1. Page 1 line 31-32. The authors write "CH4 is often also included because its lifetime of around 10 years is shorter than or comparable to climate response timescales." I am not sure if I have seen this argument before. Please, justify with a reference why this is the reason why CH4 is included among the short lived climate forcers.

In the present literature, CH4 is sometimes included as an SLCF and sometimes as a well-mixed gas. Climate & Clean Air Coalition considers CH4 to be a SLCF. We are following IPCC AR5, Myhre et al. (2013). They write: "These compounds do not accumulate in the atmosphere at decadal to centennial time scales, and so their effect on climate is predominantly in the near term following their emission." In mitigation strategies, CH4 can be treated differently as CH4 has clearly a shorter lived impact on the climate than CO2. We have changed the sentence to:

"CH4 is included in the definition because its lifetime of around 10 years is shorter than timescales for stabilizing the climate (Aamaas et al., 2016).

2. Page 4 line 131-132. The authors write "We assume that the time evoluation of temperature in each response band follows the global mean temperature". Is this a valid assumption? For example, Cherubini et al (2016) do a related, but brief analysis using MAGICC, where they estimate regional metrics based on emissions that take place in either NH-ocean, NH-land, SH-ocean and SH-land, and where the NH and SH temperature response is analysed. If one contrast the assumption that the "time evaluation of temperature in each response band follows the global mean temperature" with results presented in Cherubini et al (2016) figure 3 the assumptions appears to be rather crude, especially on short time scales. The authors need in greater length justify this assumption.

Emission metrics are a simple tool, and we want to keep it simple. MAGICC can therefore not be introduced into the methodology. We have followed what is common practice in the literature, such as by Collins et al. (2013). The short answer why we assume the same temporal response pattern is that simple and well-tested parameterizations of such temperature ratios for different latitude bands or for land vs. ocean do not exist. While we agree this is a simplification, based on Figure 3 in Cherubini et al. (2016), this simplification only gives large uncertainties in the first 5-10 years after emission. Our main case is ARTP(20), hence, this simplification plays a minor role. We have added this after the sentence:

"Cherubini et al. (2016) show that this simplification is problematic for the first 5-10 years after emissions, but leads to less uncertainty after 20 years, which is our focus."

In the section on uncertainty, we have added several paragraphs on how the temperature response may vary, such as due to the land-ocean contrast, and what new research we would like the most.

3. Equation 2. In equation 2 the indices r,m,s, for the ARTP is dropped. Why? Please, be consistent throughout the equations or explain carefully difference between similar variables used in different equations.

All the indices have now been included.

4. Equation 3 and line 14-143. The authors write: "the general expression for the ARTP can be simplified to". Even though the mathematics behind this approximation is quite simple it should be shown and/or explained in a footnote, in the supplementary material or a reference to a paper where this is done should be included.

We add a reference to Appendix 2 in Fuglestvedt et al. (2010).

5. Page 5 line 152. The authors mention that the average adjustment time of CH4 is 9.7 years in the three models used. This is relatively short compared to the IPCC AR5 assumption (12.4 years). Can the authors explain why a relatively short atmospheric adjustment time is find in the models used in the paper?

This is a good review question. Our manuscript is a follow up of the RF dataset in Bellouin et al. (2016) (see their Table 7). We have added a reference to this table in our manuscript. The adjustment time is calculated from $\tau tot*f$. As the adjustment time of CH4 is discussed in Bellouin et al. (2016), we would like to keep the discussion limited in our paper. They found a large variability in the adjustment time between models, which is to be expected and within the model diversity seen in past studies. The low average adjustment time may be due to the selection of models, particularly the inclusion of HadGEM3 with a short adjustment time. We have added this sentence:

"If we use the adjustment time of 12.4 yr from Myhre et al. (2013), the ARTP values would be larger."

6. Page 5 lines 164-165. The authors write "RCS matrices only exist for annual emissions, we assume we can apply the same set of matrices for 165 emissions during NH summer and winter." Please justify this assumption.

This is a good comment, which at present cannot be quantified, as there are no climate model simulations available that has simulated RCS coefficients for seasonal emissions. However, we still believe that there is value added through this approach. The standard annual mean ARTPs quantify the relation between a unit pulse emission and an annual mean temperature response. Applied to a specific mitigation measure (e.g. improvement in wood burning stoves used for heating to reduce BC emissions) would give a seasonal cycle in the amount of mitigation (to a varying degree depending on source). In this case, both the emission → RF and RF → response (RCS) are implicitly assumed to follow the annual mean. In our approach, we resolve the emission → RF part on a seasonal basis, but we have to keep the assumption about the RF → response part. The simple answer why we used RCS for annual emissions is that there is no alternative in the literature. New research on this is highly welcome. This issue can open for a big discussion, which we hope future research will take on. We already state that there is no alternative, but will add these sentences:

"This assumption is a simplification, but is done implicitly when the annual mean RCS are applied to seasonal varying sources, e.g., wood burning heating stoves. We believe that calculating explicitly the RF from each season improve the overall ARTP values."

7. Page 8 lines 277-279. The authors write "For all the species, the response bands with the largest ARTP values are for the responses in the NH mid-latitudes (60% of the cases) and Arctic and the band with the least response the SH mid-high latitudes (see all panels in Fig. 1). This skewness is partly due to the emissions occurring mainly in the NH, but the same pattern is seen for CH4 (Figure 1(O)), for which the emission location is less important." The argument "This skewness is partly due to the emissions occurring mainly in the NH" is confusing. I first thought that the authors were referring to actual real world emissions, but that is totally irrelevant since you study equally sized emission pulses from different regions. Please write clearly what you mean with "This skewness is partly due to the emissions occurring mainly in the NH". There is also a similar argument in line 282 where the authors write "most emissions occurring in NH". Please clarify!

The RF we have used from Bellouin et al. (2016) are based on real-world emissions. The RFs have been normalized per unit emissions, so the reviewer is correct that we in some sense are comparing equal emission pulses. But for basically all the emission regions in this study, most of the emissions of that unit of emissions occur in the NH. The point is that the RF tend to be largest near the emission source, and those emission sources are in the NH in our study. We have reformulated to:

"). This skewness towards the NH is partly due to the emissions occurring in the NH for Europe and East Asia, as well as mainly for the global emissions…"

We have also clarified in line 282 by stating that the emissions occur in the NH "for the emission regions" we looked at.

8. Figure 4 and page 13 line 445-447. The authors write "The relative differences are generally larger for the aerosols than the ozone precursors, as seen in Fig. 4, where only the emissions regions and seasons with a relative difference larger than 20% are presented." Why is only cases where the relative difference between ARTP and AGTP are larger than 20% shown? Wouldn't it be equally relevant to see the cases where the difference is small?

For presentation purposes, we select a few cases. The total number is 70. We think the most interesting is where we find the largest differences and we therefore went for those with larger differences than 20%. Some more information is given in Section 7 in Supporting Information.

References

Joshi, M.M., Gregory, J.M., Webb, M.J. et al., (2008) Mechanisms for the land/sea warming contrast exhibited by simulations of climate change. Clim Dyn 30: 455. doi:10.1007/s00382-007-0306-1

Boer, G.J. (2011) The ratio of land to ocean temperature change under global warming. Clim Dyn 37: 2253. doi:10.1007/s00382-011-1112-3

Cherubini F., J. Fuglestvedt, T. Gasser, A. Reisinger, O. Cavalett, M.A.J. Huijbregts, D.J.A. Johansson, S.V. Jørgensen, M. Raugei, G. Schivley, A. Hammer Strømman, K. Tanaka, A. Levasseur (2016) Bridging the gap between impact assessment methods and climate science. Environmental Science & Policy 64: 129-140

For other references please see the paper by Aamaas et al, 2017

---

## Author Comment (AC2) · 21 Jun 2017

We thank all the reviewers for helpful comments that will improve our manuscript. Our responses are given in red.

**Anonymous Referee #1**

The paper presents estimates of regional temperature change in latitude bands from regional emissions of several different short-lived climate forcers based on radiative forcing calculations and regional climate sensitivities from litterature. The paper is well presented and the results are useful for first order assessment of the climate impacts of different mitigation strategies for air pollutants. The paper merits publication in ACP after considering the following comments.

Section 3.4 summarizes different uncertainties. It would be useful if the authors could extend the section to include also a discussion of what research that is most urgently needed to reduce the uncertainties in the methodology used.

This is an important comment. We have added a paragraph of what research we would like to be done to improve the results:

"More research is warranted to improve the temperature estimates and to reduce uncertainties. As the forcing-response coefficients (RCS) come mainly from one model, research is most needed to test the robustness of those model results, preferably in a multimodel intercomparison framework. We would also like to encourage work on how the temporal temperature response varies between the different latitude bands and species. As new data on RF from more and smaller emission regions are published in the future, and if RCS values become available for additional forcing and response regions, our study could be extended with this improved data."

p9 l319 replace of with on

Corrected.

p10 l337 suggest "more detailed estimates for"

Accepted.

p11 l370-371 check language

We have clarified by stating "summer emissions" rather than "summer" and the same for winter. We have also changed the start of the sentence from "the temporal variability shows" to "the results show".

---

## Author Comment (AC3) · 21 Jun 2017

We thank all the reviewers for helpful comments that will improve our manuscript. Our responses are given in red.

**Anonymous Referee #2**

The manuscript by Aamaas et al. presents new calculations of regional temperature change potentials (ARTPs) calculated using radiative forcing estimates from four different models, and regional (actually zonal) climate sensitivities from one model, all taken from past studies. In a way it replicates the work of Collins et al. (2013), though with different data used for the radiative forcing estimates. It also includes some methodological advances compared to that previous work, such as the separation of the impact of different seasons, and accounting for the vertical structure of BC. The paper is certainly within the scope of ACP. It does not include any major specific new findings (those have been documented in earlier papers on which it is based), but it will be a useful addition to the literature when it comes to exploring the development and application of regional emission metrics. Therefore, I suggest its publication following the revisions and clarifications suggested below.

GENERAL COMMENTS:

- I find the title somewhat misleading. It suggests that multiple models were used, but for the actual temperature response, the calculations still rely on one model. I suggest that this kind of title is kept for when the community has RCSs from more than one model, and therefore I recommend removing that part of the title of the current manuscript.

We agree that "multiple models" might be misleading. We have readjusted the title to:

"Regional temperature change potentials for short lived climate forcers based on radiative forcing from multiple models"

- I suppose if a policy maker (or a scientist) was in need of regional metrics, they might end up being confused as to whether they should use those presented here or those presented in Collins et al. (2013). The models used for radiative forcing estimates here may be somewhat newer, but they are also fewer. Is there anything convincing that could be said (perhaps in Sect. 3.1.3) as to which of the estimates is more reliable, specifically when it comes to the radiative forcing terms? Also, it would have been interesting to see how the numbers in this study would have differed had the same method as Collins et al. (2013) been used here (i.e. without the methodological advances), but I appreciate that this may be a quite substantial task at this stage.

The reviewer is raising an interesting question. The methodological improvements and the separation between summer and winter emissions are the most important arguments for going for our study. But other factors, such as more models included, will give more weight to Collins et al. (2013). Bellouin et al. (2016) make a comparison of the RF data with previous studies (such as Yu et al. (2013) and Fry et al. (2012), which is the background to Collins et al. (2013)) in their Table 1. As a result, we are not repeating that comparison in our study. We add two sentences in Section 3.1.3, but we do not want to make a final advice on not using Collins et al. (2013):

"The study by Collins et al. (2013) is more comprehensive than our study in terms of the number of models included, while the RF dataset we use is newer and more detailed (see Table 1 in Bellouin et al.,

2016) and the forcing-response coefficients are improved. Hence results from both studies will be of benefit to those wanting to apply our metrics."

While a comparison with Collins et al. (2013) by using their methods is an excellent idea, we would like to keep that out in the interest of keeping the article length short, as well as we are uncertain how much more value that would give to our study. This would also be a substantial task at this stage, as the reviewer recognizes.

SPECIFIC COMMENTS:

Page 1, Line 14: Suggest changing "the globe" to "the entire globe".

Accepted.

Page 1, Lines 21-22: Sentence not entirely clear.

We have included "per unit BC emission" to clarify the statement:

"The temperature response in the Arctic per unit BC emission is almost 4 times larger and more than 2 times larger than the global average for Northern Hemisphere winter emissions for Europe and East Asia, respectively"

Page 1, Line 31: Suggest rephrasing to "included in the definition because".

Accepted.

Page 1, Line 32: The temporal variation has changed over time?

The word "temporal" is removed to avoid saying the same thing twice.

Page 2, Line 43: component -> constituent.

Accepted. We have also made similar changes throughout the manuscript.

Page 2, Line 54: I do not think that all the papers referenced here (e.g. Stevenson et al. (2005), Wild et al. (2001), Fry et al. (2012)) quantify the global temperature response to emissions broken down by region. But that is what the reader is left to think.

We agree that those studies did not quantify the global temperature response. We tried to be general and include both temperature and RF in the word "global response". To simplify, we will remove Berntsen et al. (2005), Stevenson et al. (2005), Wild et al. (2001), Fry et al. (2012)).

Page 2, Lines 55-57: Strictly speaking, Shindell and Faluvegi (2010) did not present regional temperature potentials; a potential is a response per unit emissions, whereas their paper provided responses per unit forcing.

We see your point. We change the end of the sentence from "to regional emissions" to "from regional RFs".

Page 2, Lines 67-68: I think you need to explain up front to the reader what the differences are with Aamaas et al. (2016).

We have changed the sentence, so there is a short description of the Aamaas et al. (2016) paper. End of sentence:

"…and extend the global temperature responses estimated by Aamaas et al. (2016) to responses on latitude bands."

Page 3, Line 73: Applied to do what?

We have added this in the sentence:

"…applied to calculate regional temperature responses of…"

Page 3, Line 82: Perhaps rephrase to "The regional RFs are then averaged for four latitude bands: : :"?

Changed.

Page 3, Line 86: Perhaps "effects" is a slightly better choice of word here than "processes". Also, I would say it is worth briefly mentioning/listing those effects here in the sentence so that the reader does not have to necessarily look at the figure to see what is meant.

Accepted. We have changed the wording throughout the manuscript. We have added the six processes/effects in the text.

Page 3, Lines 89-90: So, these are instantaneous forcings?

Page 3, Line 94: "methane induced" -> "methane-induced".

Fixed.

Page 3, Lines 100-102: Not clear how such an experiment can diagnose the semidirect effect alone. Imposing a perturbed concentration of BC would have both direct and semi-direct effects, no?

We tried to be short, but we have now expanded the explanations to:

"The semi-direct effect is quantified in Bellouin et al. (2016) by  prescribing control and perturbed distributions of BC mass-mixing ratios based on OsloCTM2 in 30-year, fixed sea-surface simulations with the Community Earth System Model (CESM). The RF from aerosol-radiation interactions was quantified with multiple calls to the radiation scheme. Because the semi-direct effect is not included in the CAM4 component of the CESM, the semi-direct effect is calculated as the difference between the RF from aerosol-radiation interactions and the effective RF."

Page 3, Lines 102-104: Not clear what ozone and methane have to do with aerosol direct and 1st indirect effects.

We have added this sentence:

"The ozone precursors and $CH_4$ can influence the aerosol effects, as a reduction in $CH_4$ concentration leads to increase in OH, which promotes sulfate aerosol formation."

Page 3, Line 108: So what is ECHAM6 used for in this study?

This is a good comment. ECHAM6 is included in the checking of robustness of individual models against the best estimate in Section 3.1.4.

Sect. 2.2: What are the implications of using RCSs derived from equilibrium simulations to infer metrics for transient situations?

This is a comment relevant for all studies on emission metrics, whether calculations are on ARTPs or AGTPs. Emission metrics are, by their nature, based on simplifications. Ideally, we would like to use temperature responses specific for each species, while what we use is the temperature response estimated by Boucher & Reddy (2008) because it has been widely used in similar work, including in IPCC AR5. See also response to reviewer 4 on temporal temperature response, which lead to including a reference to Cherubini et al. (2016).

Cherubini F., J. Fuglestvedt, T. Gasser, A. Reisinger, O. Cavalett, M.A.J. Huijbregts, D.J.A. Johansson, S.V. Jørgensen, M. Raugei, G. Schivley, A. Hammer Strømman, K. Tanaka, A. Levasseur (2016) Bridging the gap between impact assessment methods and climate science. Environmental Science & Policy 64: 129-140.

Page 4, Line 129: "Pattern" implies "geographical distribution", so I suggest replacing with e.g. "ratios".

We have replaced "pattern" with "in the different latitude bands".

Page 3, Line 138: It is not mentioned what t' represents. Presumably it is the timing of emission. Also, the indexing of location and season is suddenly dropped here. It should be mentioned in the sentence before the equation that this equation holds for the impact of every region/season or summations need to be added around the integral.

We have added information, such as stating that t is the time and included the indexing.

Page 5, Line 154: Shouldn't the upper limit of integration be H?

Yes, this is corrected. RF(H-t) has also been corrected to RF(H) in the same equation.

Page 5, Line 160: do -> apply.

Changed.

Page 5, Lines 162-163: What is meant here by "are only incorporated in RT"?

The climate sensitivity is only found in the temporal temperature response function (RT). We have change the last part of the sentence to:

"the climate sensitivity in our ARTP calculations is only included in one of the parameters, in the temporal temperature response (RT)."

Page 5, Line 166: What does CO2 have to do with scattering aerosols?

We follow the practice in the literature, the first study to take the average was Shindell & Faluvegi (2010). This was probably done to get more robust numbers and that the coefficients did not vary that much between SO2 and CO2.

Page 5, Line 167: I suggest changing "long-lived" to "longer lived", as, from a climate perspective, "long-lived" implies something even longer.

Accepted.

Page 5, Lines 165-169: I am not at all sure what has been done here. Shindell and Faluvegi (2009) provide values for every individual effect, i.e. sulphate (proxy for all scattering aerosols), ozone, BC, CO2, methane. So why haven't those simply been used here?

As above, we follow the practice by Shindell & Faluvegi (2010) and Collins et al. (2013).

Page 5, Line 170: "based on several sources" is vague here.

We have rephrased, so that part of the sentence is now this:

"the regional sensitivity matrix applied is more complex"

Page 5, Line 171: Is it really for the "aerosol effects", or for the BC part of the aerosol effects?

We have included the word BC.

Page 5, Line 173: Again, what does CO2 have to do? I may be missing something, but I guess so will several other readers, since this is not explained clearly.

We are not aware of any RCS in the literature on the semi-direct effect. Our strategy is that if we do not know better, we use the RCS values presented by Shindell and Faluvegi (2010). We think CO2 is a decent second best option. We have split the sentence into two and edited the second sentence to:

"As we are not aware of a RCS matrix for RF explicitly calculated for the semi-direct effect, we use the average CO2 and SO2 coefficients shown in Shindell and Faluvegi (2010) based on Shindell and Faluvegi (2009)."

Page 6, Line 187: do -> apply.

Changed.

Page 6, Lines 190-193: A few references are needed to support these statements.

We have added reference to Quinn et al. (2008). We think the other references later in that section covers the material.

Quinn, P. K., Bates, T. S., Baum, E., Doubleday, N., Fiore, A. M., Flanner, M., Fridlind, A., Garrett, T. J., Koch, D., Menon, S., Shindell, D., Stohl, A., and Warren, S. G.: Short-lived pollutants in the Arctic: their climate impact and possible mitigation strategies, Atmos. Chem. Phys., 8, 1723-1735, 10.5194/acp-8-1723-2008, 2008.

Page 6, Line 212: This is somewhat confusing. Since Flanner (2013) is used for the estimates of sensitivities for the case of BC on snow effects, how can the semi-direct effect be implicitly included?

Flanner (2013) is also used for the BC aerosols effects for Arctic-to-Arctic, and this sentence is based on that fact. We have included this clarification in the sentence:

"for the BC aerosol effects for Arctic-to-Arctic warming"

Page 7, Line 234: Suggest rephrasing to "Results for continuous time horizons: : :".

Changed.

Page 7, Line 242: regions -> changes.

We have changed from "the other regions" to "other differences".

Page 8, Line 252: Not clear what is meant.

We have changed sentence to:

"Due mainly to heat transport between the latitude bands, the RCS coefficients also represent non-local temperature responses, thus, the temperature response is seen more evenly in all latitude bands."

Page 8, Line 265: Is the increased efficacy a consequence of accounting for the vertical structure? If so, worth mentioning.

This line is about the efficacy of BC in snow, not the vertical structure in the atmosphere. The reviewer may have meant another line. We do not have evidence that the vertical structure gives a significant increase in efficacy for ARTP relative to AGTP. We keep sentence as is.

Page 8, Lines 273-274: (Last sentence in paragraph) On this timescale only, right? Also, this seems to also hold for NOx; perhaps worth mentioning?

CO has a longer lifetime than NOx, so this is most relevant for CO. We keep the sentence as is.

Page 8, Lines 277-278: Emissions should not matter since the metrics are normalized by emissions, right?

Yes, emission size should not matter due to normalization. But the location of the normalized emissions matter, which is the focus of this sentence.

Page 8, Lines 280-281: OK, but why? Shindell et al. (2015) provide some insight worth discussing.

Thanks for reference. We have added this sentence:

"Shindell et al. (2015) argue that the high responses in NH mid- and high-latitudes are not due to feedbacks particular for the SLCFs, but mainly due to the efficacies driven by the large land fraction in this area and strong snow albedo feedbacks."

Page 9, Line 299: Due to the largest presence of snow in the winter in the NH, presumably?

Yes. We add this to the end of the sentence:

"when the snow cover area is at its largest"

Page 9, Line 302: Why is there this negative aerosol response to VOCs?

We have added this sentence:

"VOC emissions perturb aerosols via secondary organic aerosol formation, which two out of three models find to be cooling."

Page 9, Line 308: Is it really the emissions region that drives larger differences than for other species, or the response region? It seems that the latter is the case.

We focused on different emission regions and seasons, but we see that response regions should also be mentioned here. We have added this to the sentence:

",and response regions (comparing Arctic with other latitude bands for European emissions in Fig. 1(B))."

Page 9, Line 319: of -> on

Changed.

Page 10, Line 329: for -> among

Changed.

Page 10, Line 336: Shouldn't Shindell and Faluvegi (2009) be cited here?

Yes. Included.

Page 10, Lines 343-345: Whereas what is the case here?

We extend the sentence with this:

"whereas Collins et al. (2013) did not."

Page 10, Line 360: disagree -> disagrees

Changed.

Page 11, Lines 370-371: Is this because in the summer there is more OH expected per unit NOx change, due to higher insolation?

Yes, we agree. However, as the RFs and the chemistry behind were discussed by Bellouin et al. (2016) and in the interest of space, we would like to avoid too much discussion of chemistry. We would like not to mention this detail. We see the sentence was unclear, so we have edited to:

"The results show that NOx emissions in Europe have in general more negative ARTP values for summer emissions than for winter emissions"

Page 11, Line 380: Link with the earlier Equation 2.

We have added a link to Equation 2 by attaching this to a sentence:

"based on Eq. (2)"

Page 11, Line 382: Pulse emissions, not sustained, right?

Correct, we have clarified by adding "pulse".

Page 11, Lines 388-390: Larger than what? Presumably it refers to winter vs summer, but it needs to be clarified.

Yes. We have added this clarification:

"in winter than in summer"

Page 11, Lines 390-391: All ozone precursors or just CO?

All the ozone precursors. We have added the word "all".

Page 12, Line 406: A bit confusing that earlier methane was included and here it is not, especially since often the "short-lived" terminology includes methane. Anyway, apparently methane has been dropped in this section, and it has to be mentioned upfront in it that it is not accounted for.

We have clarified stating "non-$CH_4$ SLCFs" in the first sentence.

Page 12, Line 410: Larger even than those of sulphate? I doubt it. Probably it is meant that there is a larger seasonality for BC than for other species.

Yes, we were thinking about the seasonality of BC, but we see that sulphate should also be mentioned. We have changed the sentence to:

"The main reasons for the seasonality differences are the strong heating from the BC deposition on snow for winter emissions close to snow and ice surfaces, the relatively larger BC emissions in winter than for the other species, and weaker cooling effects of SO2 in winter."

Page 13, Line 444: Perhaps add "for the same species" after "seasons".

Added.

Page 13, Line 456: Uncertainty in emissions is not accounted for, right? In which case Ei should be removed from the fraction shown in this sentence.

This is a good comment, but we think we should keep as is. The data from Bellouin et al. (2016) is normalized radiative forcing; hence, emissions are included in the denominator. The uncertainty we discuss is on the normalized radiative forcing.

Page 13, Line 469: Correlated with what?

We have clarified by adding this after correlated:

"for different species in a model"

Page 14, Line 476: And not just for SLCFs, right? Given that WMGHGs also cause regionally varying effects.

The manuscript does not address long lived greenhouse gases, but we agree that this point could be made here. We have added long lived greenhouse gases in parenthesis.

Sect. 3.4 (general): And what about the propagation of uncertainties in RCS due to internal variability, as reported in Shindell and Faluvegi (2009)? And also uncertainties due to spatial variability and subsequent averaging? Not that it would be expected to account for them at this stage, but worth mentioning and perhaps speculating on their importance.

The RTP concept (and also other emission metrics) quantifies the expected response to an emission perturbation. This is understood as the mean response for a large ensemble. For natural climate system and for a single GCM simulation there will be unforced natural variability on top of that. The uncertainty numbers in Shindell and Faluvegi (2009) are the standard deviation of the last 80 years of an equilibrium

run. As such, it is only a measure of the unforced internal variability. Since this should not be a part of the metric values, we believe it is not correct to include it in the analysis.

We have added several paragraphs in Section 3.4 on uncertainties related to spatial variability as a response to other review comments. See the other reviews.

Page 14, Line 487: Suggest removing "the more".

Deleted.

Page 14, Line 491: indicate -> indicates

Changed.

Page 14, Line 500: Suggest adding "by" before "up" and "individual" before "regions".

Added.

REFERENCES:

Shindell, D. T., G. Faluvegi, L. Rotstayn, and G. Milly (2015), Spatial patterns of radiative forcing and surface temperature response, J. Geophys. Res. Atmos., 120, 5385–5403, doi:10.1002/2014JD022752.

---

## Author Comment (AC4) · 21 Jun 2017

We thank all the reviewers for helpful comments that will improve our manuscript. Our responses are given in red.

**D. Henze**

daven.henze@colorado.edu

The manuscript by Aamaas et al. (2017) evaluates the impact of emissions changes resolved across four latitudinal bands on temperature change. Overall, this is a very valuable work, and we appreciate several aspects of this paper, especially the multimodel efforts and exploration of the role of altitude for BC. However, discussion of the works cited in this comment and associated caveats on the use of regional temperature potentials should be included.

The use of latitudinal bands stems from the work of Shindell and Faluvegi (2009) and Shindell (2012), which evaluated climate response to radiative forcing in these bands. However, we take issue with the application of these bands for defining relationships between emissions and radiative forcing, which ignores previously published work quantifying variability of radiative forcing efficiencies within these bands of more than an order of magnitude.

We have chosen to use the RTP-concept for our temperature calculations and since the Shindell sensitivities are the only ones currently available to us, we have to use those bands. As noted in our response to other reviewers, it would be highly desirable to have more refined calculations in the future. Our approach and focus could of course have been different. In the interest of space we have limited what we have referred to and discussed. However, we see that other aspects are relevant and we include some text and some of the references given, see answers below.

As noted by another reviewer, these regions are presented in Aamaas et al. (2017) arbitrarily in the context of relating emissions to radiative forcing, without consideration of more highly resolved regions, or – even more importantly – resolving emissions at scales other than latitudinal bands.

We might be misunderstanding this comment, as we did not look at emissions from latitudinal bands, but from regions such as Europe. The RFs were estimated by Bellouin et al. (2016), and most of the review comments would have been more suitable for that paper. See response to reviewer 4 on why we do not have higher resolution for the response than four latitudinal bands. While it would be interesting to look at emissions from smaller scales (such as what was done in Henze et al. (2012) and Bowman & Henze (2012)), we decided to investigate emissions for large regions. An improvement in our study is that we separate between summer and winter emissions.

Further, we have in our own work explicitly shown that radiative forcing efficiencies of aerosol (Henze et al., 2012; Lacey et al., 2015) and ozone (Bowman and Henze, 2012) precursor emissions vary tremendously – more than 1000% – across latitudes. For aerosols, the key features modulating radiative forcing efficiency are related to aerosol lifetime over surfaces of varying albedo and the chemical environment for forming secondary PM from gas-phase precursors (such as the ratio of ammonia to sulfate and nitric acid). Latitude has little bearing on aerosol radiative forcing efficiency, although (the Himalayan region aside) it does impact the indirect effects of BC deposition on snow and ice (Lacey et al., 2015). For short term ozone direct radiative forcing (DRF) efficiency, latitude is a key variable, but

also factors such as atmospheric chemistry, altitude, and the efficiency of vertical mixing play important roles. For example, Bowman and Henze (2012) find that "NOx emissions in Chicago would lead to 0.01 mW/m2 change in DRF but the equivalent absolute reduction to emissions east of Atlanta would lead to a 0.035 mW/m2 DRF reduction."

We add a sentence in the introduction (the second paragraph) on the finer scales:

"While we focus on RF from large emission regions, Bowman and Henze (2012);Henze et al. (2012) showed that radiative forcing efficiencies can vary by 1000 % for much smaller emission regions."

We have also added a paragraph in the uncertainty section:

"The ARTP values are given for large emission regions, while large variations are likely within the regions. The impact of emissions from an European city may be very different to the average we have estimated for European emissions (see Bowman and Henze, 2012;Henze et al., 2012). They found that the key determinants for aerosols are the aerosol lifetime, surface albedo, and the chemical environment. Latitude is a key variable for ozone, but atmospheric chemistry, altitude, and vertical mixing play also a role."

We hope that a revised manuscript from Aamass et al. will consider these important factors in their presentation and evaluation of regional temperature potentials, explicitly stating the uncertainties in application of their coefficients to evaluate temperature impacts of emissions changes at scales other than latitudinal bins. For example, while the latitudinal dependency of climate sensitivities imparts a strong-latitudinal dependence on the overall regional temperature potentials for emissions, the sub-latitudinal impact of emissions on radiative forcing can lead to important differences in the temperature impacts of equivalent changes to emissions in country-scale cookstove mitigation sce-narios (Lacey and Henze, 2015; Lacey et al., 2017).

We have added some more text on uncertainty based on our response to other reviewers. But we have not calculated temperature impacts based on emissions in latitude bins. We have calculated temperature due to emissions from regions, which is very similar to Lacey and Henze (2015) & Lacey et al. (2017), just at a cruder scale.

Daven K. Henze, Associate Professor, Dept of Mechanical Engineering, University of Colorado at Boulder. Boulder, CO, USA.

Forrest Lacey, Ph.D., Postdoctoral Researcher. National Center for Atmospheric Research (NCAR), Boulder, CO, USA.

Bowman, K. W., and D. K. Henze (2012), Attribution of direct ozone radiative forcing to spatially-resolved emissions, Geophys. Res. Lett., 39, L22704, doi:10.1029/2012GL053274.

Henze, D. K., D. T. Shindell, F. Akhtar, R. J. D. Spurr, R. W. Pinder, D. Loughlin, M. Kopacz, K. Singh, and C. Shim (2012), Spatially refined aerosol direct radiative forcing efficiencies, Environ. Sci. Technol., 46, 9511 - 9518, dx.doi.org/10.1021/es301993s.

Lacey, F., and D. K. Henze (2015), Global climate impacts of country-level primary carbonaceous aerosol from solid-fuel cookstove emissions, Environ. Res. Lett., 10, 114003, doi:10.1088/1748-9326/10/11/114003.

Lacey, F., D. K. Henze, C. Lee, A. van Donkelaar, R. V. Martin (2017), Transient climate and ambient health impacts due to national solid fuel cookstove emissions, Proc. Nat. Acad. Soc., doi:10.1073/pnas.1612430114.

---

## Author Comment (AC5) · 21 Jun 2017

**Regional temperature change potentials for short lived climate forcers **based on radiative forcing** from multiple models**

Borgar Aamaas[1], Terje K. Berntsen[1,2], Jan S. Fuglestvedt[1], Keith P. Shine[3], William J. Collins[3]

[1]CICERO Center for International Climate Research, PB 1129 Blindern, 0318 Oslo, Norway

[2]Department of Geosciences, University of Oslo, Norway

[3]Department of Meteorology, University of Reading, Reading RG6 6BB, United Kingdom

*Correspondence to*: Borgar Aamaas (borgar.aamaas@cicero.oslo.no)

**Abstract.** We calculate the absolute regional temperature change potential (ARTP) of various short lived climate forcers (SLCFs) based on detailed radiative forcing (RF) calculations from four different models. The temperature response has been estimated for four latitude bands (90-28° S, 28° S-28° N, 28-60° N, and 60-90° N). The regional pattern in climate response not only depends on the relationship between RF and surface temperature, but also on where and when emissions occurred and atmospheric transport, chemistry, interaction with clouds, and deposition. We present four emissions cases covering Europe, East Asia, the global shipping sector, and the entire globe. Our study is the first to estimate ARTP values for emissions during Northern Hemisphere summer (May-October) and winter season (November-April). The species studied are aerosols and aerosol precursors (black carbon (BC), organic carbon (OC), $SO_2$, $NH_3$), ozone precursors ($NO_x$, CO, volatile organic compound (VOC)), and methane ($CH_4$). For the response to BC in the Arctic, we take into account the vertical structure of the RF in the atmosphere, and an enhanced climate efficacy for BC deposition on snow. Of all SLCFs, BC is the most sensitive to where and when the emissions occur, as well as giving the largest difference in response between the latitude bands. The temperature response in the Arctic per unit BC emission is almost 4 times larger and more than 2 times larger than the global average for Northern Hemisphere winter emissions for Europe and East Asia, respectively. The latitudinal breakdown gives likely a better estimate of the global temperature response as it accounts for varying efficacies with latitude. An annual pulse of non-methane SLCFs emissions globally (representative of 2008) leads to a global cooling. Whereas, winter emissions in Europe and East Asia give a net warming in the Arctic due to significant warming from BC deposition on snow.

**1 Introduction**

Climate is influenced by a multitude of emissions with varying impacts (e.g., Myhre et al., 2013). Emissions of short lived climate forcers (SLCFs), such as black carbon (BC), organic carbon (OC), $SO_2$, $NH_3$, $NO_x$, CO, and volatile organic compounds (VOCs), affect the composition of the atmosphere primarily on time scales of days to a few months. $CH_4$ is included in the definition  because its lifetime of around 10 years is shorter than  timescales for stabilizing the climate (Aamaas et al., 2016). The  variation in the geographical pattern of SLCF emissions has changed over time, with emissions typically being high in the early phases of industrialization, and then gradually being reduced due to air quality concerns and technological improvements. Nevertheless, emissions are still growing in many parts of the world, and there is a
growing focus politically to develop mitigation strategy for the SLCFs to achieve both improved air quality and
slowing global warming (Schmale et al., 2014;Shindell et al., 2012;Stohl et al., 2015).

Due to the short atmospheric lifetimes, emissions of SLCFs lead to a spatial pattern in radiative forcing (RF) that
is more inhomogeneous than for emissions of long-lived greenhouse gases such as $CO_2$. While we focus on RF
from large emission regions, Bowman and Henze (2012);Henze et al. (2012) showed that radiative forcing
efficiencies can vary by 1000 % for much smaller emission regions. It is well established that there is not a close
relationship between the RF pattern and the surface temperature response pattern, due to modifications by heat
transport in the atmosphere and ocean and the spatial variability in climate feedbacks (e.g., Boer and Yu, 2003).
However, as shown by Shindell and Faluvegi (2009) and Shindell (2012), it is possible to establish relationships
between the RF pattern caused by a certain constituent and the response in broad latitude bands.
Recently, Najafi et al. (2015), have shown from observational and model data that there is a distinct difference in
the Arctic response to the overall forcing by ozone, aerosols and land-use, compared to other latitude bands.

Emission metrics are simple tools based on comprehensive model simulations that relate emissions to a certain
response (physical climate change or economic damage), e.g. Fuglestvedt et al. (2003);Tol et al. (2012). The most
widely used emission metric, the Global Warming Potential (GWP), is given by the integrated RF (over a time
horizon of H years) in response to a pulse emission. Shine et al. (2005) introduced the Global Temperature change
Potential (GTP), using the surface temperature change (after a time horizon of H years) for the response. Emissions
metrics have typically estimated a global effect due to global emissions (e.g., Aamaas et al., 2013). A first step
going beyond global means was to quantify the global response based on regional emissions for SLCFs
(Fuglestvedt et al., 2010;Collins et al., 2013;Aamaas et al., 2016). By introducing the concept of regional
temperature potentials (RTP), Shindell and Faluvegi (2010) extended the metric concept to include regional
responses (in terms of surface temperature change in broad latitude bands) from regional RFs.

In addition to the regionality, the timing of the SLCFs emissions matter. This is potentially important since the
photochemistry in the atmosphere, lifetime, atmospheric transport and forcing efficiency is likely to vary between
the seasons. As some sources (e.g. domestic heating and agricultural waste burning) have a large seasonal cycle,
using seasonal RTP metrics might have a significant impact on the evaluation of cost-effectiveness of mitigation
measures.

Here we use detailed multimodel calculations of the relationship between emission location and the resulting
specific RF (RF per Tg/yr emissions) for SLCFs (Bellouin et al., 2016) (Sect. 2.1) and the regional climate
sensitivities (e.g., Shindell and Faluvegi, 2009) to estimate ARTPs for a range of aerosols, aerosol precursors, and
ozone precursors (BC, OC, $SO_2$, $NH_3$, $NO_x$, CO, and VOC), and $CH_4$ (Sect. 2.2). The findings
mostly confirm the results by Collins et al. (2013 and extend the global temperature responses
estimated by  Aamaas et al. (2016) to responses on latitude bands. Our study is
the first to calculate ARTPs for $NH_3$ emissions. The treatment of BC in the Arctic is more complex which has a
high influence on the ARTPs for BC. Aspects of the aerosol effects on ozone precursors are also novel. For the
first time, we distinguish between ARTPs for emissions taking place during Northern Hemisphere (NH) summer
(May-October) and winter (November-April). ARTP metrics are calculated for regional emissions from Europe,

East Asia and the shipping sector, as well as global emissions (Sect. 3.1). The ARTP values are applied to calculate regional temperature responses of global emissions in Sect. 3.2. We also make a comparison of ARTPs with

AGTPs (Sect. 3.3). Uncertainties are discussed in Sect. 3.4, and we conclude in Sect. 4.

**2    Material and methods**

**2.1  Radiative forcing**

The RFs that are the basis for the ARTP calculations of the SLCFs are calculated using 4 different chemistry climate models or chemical-transport models presented by Bellouin et al. (2016); see details about the models in

Table 1. RFs are produced based on a control simulation and numerous perturbation simulations that consider a

20% emission reduction in one type of species and one region in NH summer or winter. The ECLIPSE emission dataset applied here was created with the GAINS (Greenhouse gas-Air pollution Interactions and Synergies) model, see Stohl et al. (2015). The regional RFs are then averaged for four latitude bands, southern mid-high latitudes (90-28° S), the Tropics (28° S-28° N), northern mid-latitudes (28-

60° N), and the Arctic (60-90° N), as forcing-response coefficients are only available for those latitude bands in the literature (e.g., Shindell and Faluvegi, 2010;Shindell, 2012).

We compute ARTPs for six different effects that contribute to the RF for each species (aerosol effects,

BC deposition on snow, BC semi-direct, short-lived ozone, methane, and methane-induced ozone

). The quantification of these effects are given by the RF data from Bellouin et al. (2016). For the general circulation models, the RFs of the aerosol perturbations are calculated online using two calls to the radiation scheme. This method involves diagnosing radiative fluxes with and without the perturbation. These RFs do not include rapid adjustments (even in the stratosphere). For the OsloCTM2 chemistry transport model and the

RF exerted by the ozone precursors in all the models, RF is computed by offline radiative transfer codes. The RF

for methane is based on the analytical expression that includes stratospheric adjustments (Myhre et al., 1998), which gives a global mean. Based on this global RF estimate, we apply the latitudinal pattern in RF for methane and methane-induced ozone response in Collins et al. (2013). This pattern is based on an ensemble of 11 global chemical transport models that evaluated a global reduction of $CH_4$ mixing ratio, where RF was calculated using the method developed by the NOAA Geophysical Fluid Dynamics Laboratory (Fry et al., 2012).

For aerosols and aerosol precursors, all four models calculate the aerosol direct and 1st indirect (cloud-albedo)

effect, except ECHAM6 which only includes direct RF. In this study, we group together the aerosol direct and 1st indirect (cloud-albedo) effect and name this  *aerosol effects*. In addition, OsloCTM2 estimated the RF from

BC deposition on snow and the semi-direct effect. The semi-direct effect is quantified in Bellouin et al. (2016) by prescribing control and perturbed distributions of BC mass-mixing ratios based on OsloCTM2 in 30-year, fixed sea-surface simulations with the Community Earth System Model (CESM). The RF from aerosol-radiation interactions was quantified with multiple calls to the radiation scheme. Because the semi-direct effect is not included in the CAM4 component of the CESM, the semi-direct effect is calculated as the difference between the

RF from aerosol-radiation interactions and the effective RF

. For the ozone precursors and $CH_4$, the total RF takes into account the aerosol direct and 1st indirect effects, short-lived ozone effect, methane effect, and methane-induced ozone effect. The ozone precursors and $CH_4$ can influence the aerosol effects, as a reduction in

CH$_4$ concentration leads to increase in OH, which promotes sulfate aerosol formations. Only OsloCTM2 includes an estimate for nitrate aerosols, which is added to the *aerosol effect* quantification in the other models.

The best estimate of a species' RF has been calculated as the sum of all effectsprocesses, in which the average across the models is used for each effectprocess. Not all models have estimated RFs for all species and effectsprocesses. In addition, ECHAM6 is excluded in the best estimate for BC, OC, and SO$_2$, since it did not estimate the 1$^{st}$ indirect effect. For BC deposition on snow, the BC semi-direct effect, and nitrate aerosol, the best estimate is solely based on the OsloCTM2 model, while the best estimate are based on three models for all other effectsprocesses (*aerosol effects*, short-lived ozone, methane, and methane-induced ozone).

For the high and low estimates of RF for each emission case, we find these values by taking the sum of the highest and lowest values, respectively, from all models for each individual effectprocess.

The emission regions are defined according to tier1 Hemispheric Transport of Air Pollution (HTAP) regions (see

Bellouin et al., 2016). Europe is defined as Western and Eastern Europe up to 66°N including Turkey. East Asia includes China, Korea, and Japan. Shipping is the global shipping sector. The global emissions category excludes this shipping activity. As RF values are also available for the remaining land areas outside of Europe and East

Asia, results from the rest of the World are presented in SI Sect. 2.

**2.2 Regional temperature change potentials**

The regional temperature response has been calculated on the basis of RF in the latitude bands and regional climate sensitivities, as well as the temporal evolution of an idealized temperature response. Even though our estimates are based on seasonal emissions, the temperature responses calculated are annual means. The general expression for the ARTP following a pulse emission of constituentcomponent $i$ ($E_i$) in region $r$ which leads to a response in latitude band $m$ is (e.g., Collins et al., 2013):

$$ARTP_{i,r,m,s}(H) = \sum_l \int_0^H \frac{F_{l,i,r,s}(t)}{E_{i,r,s}} \times RCS_{i,l,m} \times R_T(H-t)dt \qquad\qquad (1)$$

$F_{l,r,s}$ $(t)$ is the RF in latitude band $l$ due to emission in region $r$ in season $s$ as a function of time (t) after the pulse emission $E_{r,s}$ (in Tg). Our study separates between four latitude response bands, in line with the typical width of response bands to inhomogeneous forcing found by Shindell et al. (2010), while more detailed modelling will be possible with a finer-masked RCS matrix available. The $RCS_{i,l,m}$ is a matrix of regional response coefficients based on the RTP concept (unitless, cf. Collins et al., 2013). As these response coefficients are here normalized, they contain no information on climate sensitivity, only the relative regional responses in the different latitude bandspattern. The global climate sensitivity is included in the impulse response function $R_T$, which is a temporal temperature response to an instantaneous unit pulse of RF (in K/(Wm$^{-2}$)). We assume that the time evolution of temperature in each response band follows the global-mean time evolution. Cherubini et al. (2016) show that this simplification is problematic for the first 5-10 years after emissions, but leads to less uncertainty after 20 years, which is our focus. We base our temperature response on that of the HadCM3 climate model (Boucher and Reddy,

2008) with an equilibrium climate sensitivity of 1.06 K/(W m$^{-2}$), which translates to a 3.9 K warming for a doubling of CO$_2$ concentration. This is the same climate sensitivity as for our absolute Global Temperature change Potential (AGTP) calculations on the same RF dataset (Aamaas et al., 2016).

Regional temperature responses at time $t$ of an emission scenario $E(t)$ can be calculated with these ARTP values by a convolution (see also Aamaas et al., 2016). The temperature response is:

$$\Delta T_{i,r,m,s}(t) = \int_0^t E_{i,r,s}(t') \times ARTP_{i,r,m,s}(t - t')dt' \hspace{3cm} (2)$$

**2.2.1    For species with lifetimes less than one year**

For SLCFs with atmospheric lifetimes (or indirect effects causing RF) much shorter than both the time horizon of the ARTP and the response time of the climate system (given by the time constants in $R_T$ above), the general expression for the ARTP can be simplified to (see Appendix 2 in Fuglestvedt et al., 2010):

$$ARTP_{i,r,m,s}(H) = \sum_l \frac{F_{l,i,r,s}}{E_{i,r,s}} \times RCS_{i,l,m} \times R_T(H) \hspace{3cm} (3)$$

$F_{l,r,s}$ is the RF over a year where emissions of constituent $i$ ($E_{i,r}$s in Tg/yr) in emission region $r$ occur during season $s$, either during NH summer or winter.

**2.2.2    For species that affect methane**

Methane has an adjustment time comparable to the time horizon of the ARTP and the response time of the climate system. So, for species that affect methane, an additional impulse response function that describes the atmospheric decay of methane must be included ($R_F$). In this case, we add such a function, which governs the methane and methane-induced ozone effects for the ozone precursors (NOx, CO, and VOC) and $CH_4$.

$$R_F(t) = e^{-t/\tau}, \hspace{5cm} (4)$$

where $\tau$=9.7 yr is the average adjustment time for methane in the three models (see Table 7 in Bellouin et al.,

2016). If we use the adjustment time of 12.4 yr from Myhre et al. (2013), the ARTP values would be larger. For these species, this additional temperature perturbation due to these effects has to be included:

$$ARTP(R_F \ response)_{i,r,m,s}(H) = \sum_l \int_o^{Ht} \frac{F_{l,i,r,s}}{E_{i,r,s}} \times R_F(H\text{—}t) \times RCS_{i,l,m} \times R_T(H - t)dt$$

(5)

**2.2.3    Forcing-response coefficients**

The unitless regional sensitivity matrix ($RCS_{i,l,m}$) is estimated based on literature values of regional response coefficients in K/(W m$^{-2}$) (see Sect. 1 in Supporting Information for tabulated coefficients). All these response coefficients from the different literature sources have been normalized to the global response in those studies.

While the specific regional response coefficients have been estimated in other studies based on climate sensitivities, the normalization to the global response removes the implicit climate sensitivities in the RCS values. We apply several adjustments and refinements of the RCS values (see this section and Sect. 2.2.4); in each case, we normalize the response coefficients and make sure that the climate sensitivity in our ARTP calculations is only included in one of the parameters, in the temporal temperature response  ($R_T$).

As such, RCS matrices only exist for annual emissions, we assume we can apply the same set of matrices for emissions during NH summer and winter. This assumption is a simplification, but is done implicitly when the annual mean RCS are applied to seasonal varying sources, e.g., wood burning heating stoves. We believe that calculating explicitly the RF from each season improve the overall ARTP values. For the scattering aerosols and aerosol precursors ($SO_2$, OC, $NH_3$), we use the coefficients tabulated in Shindell and Faluvegi (2010), which are the mean responses of $CO_2$ and $SO_2$. The same values are used for the longer- lived effects (methane and methane- induced ozone) of the ozone precursors and $CH_4$. For the short lived effects of the ozone precursors and $CH_4$, we apply the $O_3$ coefficients in Shindell and Faluvegi (2010) as tabulated in Collins et al. (2013).

For BC, the regional sensitivity matrix applied is more complexis based on several sources, and the details for the

Arctic-to-Arctic responses are described in Sect. 2.2.4. For other latitude bands, the matrix for the BC *aerosol*

*effects* is given by BC forcing-response coefficients from Shindell and Faluvegi (2009) as tabulated in Table 3 in

Collins et al. (2013). As we are not aware of a RCS matrix for RF explicitly calculated for the semi-direct effect, we use 
[revised manuscript text omitted]
 mainly to heat transport between the latitude bands, the RCS coefficients also represent non-local temperature responses, thus, the temperature response is seen more evenly in all latitude bandsDue to heat transport between the latitude bands and the temperature response lasting over several years, the forcing response is averaged out over several latitude bands by the temperature response. Nevertheless, the temperature response has higher sensitivity towards the Arctic and NH mid-latitude bands (see all panels in Fig.

1) as a result of local feedback processes being stronger in the Arctic, driven by local cloud, water vapor, and surface albedo feedbacks (Boer and Yu, 2003).

We next consider differences between the emission regions Europe and East Asia. The RF per unit emission is dependent on where the emissions occur, which causes differences in the ARTP(20) values. The differences in the global average of RFs and global emission metric values such as AGTP(20) are discussed in Aamaas et al. (2016).

In short, the emission metric values for the aerosols are larger for European than East Asian emissions, but not for

$NH_3$ in winter. Variations are also seen for the ozone precursors, but these differences are relatively smaller between European than East Asian emissions for CO and VOC than for the aerosols. For CO, East Asia has marginally larger values (see Figs. 1(K) and 1(L)) and marginally larger for European VOC emissions (see Figs.

1(M) and 1(N)). The main difference in the global average of ARTP values calculated here and the AGTP values calculated in Aamaas et al. (2016) is the much larger impact for BC deposition on snow for ARTP (see Fig. 1(B)), as the AGTP study did not account for the increased efficacy of BC deposition on snow.

The timing of emissions also influences the RF per unit emissions. The emission metric values for the aerosol emissions in Europe and East Asia (see Figs. 1(A)-1(F)) are larger for summer than winter, except for BC. For the aerosols, the aerosol RF is driven by seasonal variations in the incoming solar radiation. More sunlight in local summer results in stronger RFs (Bellouin et al., 2016). Seasonal differences in atmospheric lifetimes due to seasonality in precipitation may also contribute. BC is discussed in detail in Sect. 3.1.2.

For the ozone precursors (see Figs. 1(I)-1(N)), the largest values occur in winter for CO (Figure 1(L)) and in summer for VOC (Figure 1(M)). CO has a longer lifetime during local winter leading to a larger fraction of the

CO emitted being transported from the higher latitudes to the Tropics. Here, the effects of CO-oxidation on tropical

OH have the largest impacts on the methane lifetime.

The latitudinal response patterns are similar for the different species. For all the species, the response bands with the largest ARTP values are for the responses in the NH mid-latitudes (60% of the cases) and Arctic and the band with the least response the SH mid-high latitudes (see all panels in Fig. 1). This skewness towards the NH is partly due to the emissions occurring in the NH for Europe and East Asia, as well as mainly for the global emissionsin the NH, but the same pattern is seen for CH$_4$ (Figure 1(O)), for which the emission location is less important.

Further, the high ARTP values for the Arctic are also due to stronger local feedback processes, leading to larger forcing-response sensitivities, while high ARTP values for the NH mid-latitudes are a combination of high RF

values per unit emission and relatively large regional climate sensitivities. Shindell et al. (2015) argue that the high responses in NH mid- and high-latitudes are not due to feedbacks particular for the SLCFs, but mainly due to the efficacies driven by the large land fraction in this area and strong snow albedo feedbacks. The low ARTP values for SH mid-high latitudes is caused by a combination of most emissions occurring in NH for the emission regions and weaker forcing-response coefficients in SH. Let us consider OC emissions in East Asia during summer as an example with RF mostly in one band. The RF   (see Bellouin et al., 2016) in the NH mid-latitude band is 260%

above the global average, practically zero in the SH mid-high latitude band and about 50% below the global average in the other two bands. This skewedness is also modeled in the ARTP (see Fig. 1(C)), but with more emphasis on the Arctic. The ARTP value for the responses in the Arctic and NH mid-latitudes is about 70% and

90% above the global average, respectively.  In the SH mid-high latitudes response band, the ARTP value is about

20% of the global average. At the other end of the range, emissions of CH$_4$ have a global impact due to the atmospheric lifetime of CH$_4$ (9.7 years). The RF in the Arctic band is 35% below the global average, while 25%

above in the Tropics. But the weighing is almost opposite for the ARTP, as the Arctic response band has a ARTP

value 34% above the global average and the Tropics 13% above the average (see Fig. 1(O)). For the SH mid-high latitude response band, both the RF and ARTP are lower than the global average, by -35% and -49%, respectively.

For most of the aerosol emissions (see Figs. 1(A)-1(F)), the ARTP values for the *aerosol effects* component are larger for emissions in NH summer than winter, even in the Tropics for emission from both Europe and East Asia.

The only exception is NH$_3$ (Figures 1(G) and 1(H)), which has a larger ARTP value for winter than summer for

East Asian and global emissions. Longer sunlight duration in the summer hemisphere yields stronger RFs (Bellouin et al., 2016), which impact the ARTP value for the response even in the Tropics. This general observation does not hold for BC when we include the effectprocess "BC deposition on snow", as this effectprocess is largest in NH

winter when the snow cover area is at its largest.

The ARTP(20) values shift sign for some of the latitude response bands. VOC emissions generally lead to a warming, however, our best estimate indicates a small cooling in SH mid-high latitudes for European and East

Asian winter emissions (Figure 1(N)). The negative RF for the *aerosol effect* in this response band is driving this cooling as the other perturbations have a small impact on the response in the SH mid-high latitudes. VOC emissions perturb aerosols via secondary organic aerosol formation, which two out of three models find to be cooling. For the ozone precursors, the *aerosol effects*, and the short-lived ozone effect to a smaller degree, also shift between warming and cooling depending on the latitude response band.

**3.1.2    Variation of BC response with emission season and region**

The largest differences in ARTP(20) values are seen for BC, such as the timing of emissions (comparing Figs. 1(A)

and 1(B)) , the location during winter (comparing the different emission regions in Fig. 1(B)), and response regions (comparing Arctic with other latitude bands for European emissions in Fig. 1(B)).

The total emission metric values of BC emissions depend on which effects are included. The direct

*aerosol effect* is larger for summer than winter emissions. The direct temperature response is similar for emissions occurring in Europe, East Asia, and globally. Similarly, the semi-direct effect is most pronounced in summer as this effect is driven by absorption of shortwave radiation. When the influence from the BC deposition on snow is included, the ARTP value increases significantly for emissions during NH winter. For emissions in Europe, the global temperature response to the semi-direct effect is -46% and -12% of the *aerosol effect* in summer and winter, respectively, and the deposition on snow effect 12% and 230% of the *aerosol effect* in summer and winter, respectively. The relative share of the deposition on snow effect is 60 % lower for winter East Asian emissions than for winter European emissions. The semi-direct effect has a relative weight of -56% compared to the *aerosol*

*effect* for the global ARTP(20) East Asian emissions in summer and close to zero in winter. The impact of BC

deposition on snow is largest when large snow and ice covered surface areas and solar radiation at the BC

deposition location is combined, such as in late winter. The response from European emissions is larger than for

East Asian emissions since the emission region is closer to the Arctic, which makes BC transport into the sensitive

Arctic more likely (Sand et al., 2013). The effect of the BC deposition on snow dominates the winter-summer difference for BC and hence our results are sensitive to both the calculated RF and efficacy for this BC process.

The Arctic response amplification, i.e., how much stronger the response is in the Arctic relative to the global average, is largest for winter emissions as the deposition on snow effect is relatively larger than for summer emissions. The total Arctic response amplification for BC is for European emissions 240% and 390% larger than the global average in summer and winter, respectively, and for East Asian emissions 160% and 240% larger than the global average in summer and winter, respectively. As a result, wintertime BC emissions have the largest latitudinal variation in the ARTP(20) among all SLCFs. This Arctic amplification is driven by the temperature response from deposition on snow effect (almost 500% for European emissions and 400% for East Asian emissions for this effect), which is largest in the Arctic response band, above the global average in the NH mid- latitude, and below average in the two other response bands. Latitudinal response variations are also found for the other effects, but relatively much smaller.

**3.1.3    Comparison with Collins et al. (2013)**

Our findings are largely consistent with those by Collins et al. (2013). Similarities occur because the two studies share some of the same forcing-response coefficients (Shindell and Faluvegi, 2009) and climate sensitivity (Boucher and Reddy, 2008). In this work, we have more detailed estimates for BC in the Arctic and we include $NH_3$, as well as more detailed for aerosol impacts on ozone precursors. ARTP values are also given for two seasons, for the shipping sector and our global estimate include all emissions. The study by

Collins et al. (2013) is more comprehensive than our study in terms of the number of models included, while the

RF dataset we use is newer and more detailed (see Table 1 in Bellouin et al., 2016) and the forcing-response
coefficients are improved. Hence, results from both studies will be of benefit to those wanting to apply our metrics.

[revised manuscript text omitted]

The temperature response will vary by species and location, such as land surface versus ocean surface. These
differences are not accounted for in our study, but the increased efficacy in the RCS matrix towards the NH can
be partly attributed to larger land area fraction in the NH (Shindell et al., 2015). The temperature increase is in
general larger over land than ocean (Boer, 2011) driven by several local feedbacks (Joshi et al., 2008). We do not
have data to break down this effect for our emission regions, but results in Shindell (2012) indicate that the land
response may be 20 % larger than the average.

More research is warranted to improve the temperature estimates and to reduce uncertainties. As the forcing-
response coefficients (RCS) come mainly from one model, research is most needed to test the robustness of those
model results, preferably in a multimodel intercomparison framework. We would also like to encourage work on
how the temporal temperature response varies between the different latitude bands and species. As new data on
RF from more and smaller emission regions are published in the future, and if RCS values become available for
additional forcing and response regions, our study could be extended with this improved data.

The ARTP values are given for large emission regions, while large variations are likely within the regions. The
impact of emissions from an European city may be very different to the average we have estimated for European
emissions (see Bowman and Henze, 2012;Henze et al., 2012). They found that the key determinants for aerosols
are the aerosol lifetime, surface albedo, and the chemical environment. Latitude is a key variable for ozone, but
atmospheric chemistry, altitude, and vertical mixing play also a role.

Ideally, calculations of the temperature response of changed emissions of SLCFs should use earth system models for the most correct estimates. However, this is extremely time consuming, and many emission perturbations will have small signal/noise ratios. Users of emission metrics, such as policymakers and decision makers, might not have the needed expertise to utilize advanced models. Although the ARTP calculations are simplifications and contain uncertainties, these emission metrics are useful, simple, and quick approximations for calculating the temperature response in the different latitude bands for emissions of single species or a mix of SLCFs (and long lived greenhouse gases).

**4 Conclusion**

We have presented ARTP values in four latitude bands (90-28° S, 28° S-28° N, 28-60° N, and 60-90° N) for several SLCFs (BC, OC, $SO_2$, $NH_3$, $NO_x$, CO, VOC, and $CH_4$) based on four different models. Numbers are provided for emission occurring in Europe, East Asia, from the global shipping sector, as well as globally.

Emissions were separated between the NH summer and winter seasons. Although ARTPs are simplifications, they are useful for analyzing the temperature response to possible mitigation strategies. The ARTP values are largest in the response bands Arctic and NH mid-latitudes and the smallest in the SH mid-high latitudes. The different models agree in most of the cases on the ranking of the temperature perturbation in the different latitude bands.

BC is the species that is the most sensitive to the timing of emissions, to the location during winter, as well as having the largest spread in responses between the latitude response bands in winter. The relative difference between the response bands is largest for BC emissions during NH winter, and the more the closer to the Arctic the emissions occur. The Arctic temperature response is 390% and 240% larger than the global temperature response for winter emissions in Europe and East Asia, respectively. BC deposition on snow is the most important effectprocess influencing the Arctic for BC emissions occurring in NH winter, both in absolute and relative terms.

We have also investigated how the global response based on ARTP compares with AGTP. Our study indicates that the global temperature response can be better quantified with ARTPs than AGTPs since ARTPs include a simple representation of varying efficacies due to heterogeneous RFs. For global emissions of SLCFs excluding

$CH_4$, calculations based on ARTP values give 13% larger cooling than based on AGTP values. Globally, both these calculations based on ARTP(20) and AGTP(20) show a cooling, while European and East Asian winter emissions give a small net warming or near zero impact according to ARTP. This is driven by net warming in the

Arctic and close to zero perturbation in the other latitude bands. For summer emissions, net cooling occurs in all latitude bands, but are largest in the NH mid-latitudes and Arctic. Seasonal emissions and seasonal ARTP values give almost the same total temperature response as annual emissions and annual ARTP values for global emissions, but changes the temperature responses by up to 18% when looking at emissions from individual regions such as

Europe and East Asia.

**Acknowledgements**

The authors would like to acknowledge the support from the European Union Seventh Framework Programme (FP7/2007-2013) under grant agreement no 282688 – ECLIPSE, as well as funding by the Norwegian Research

Council within the project "the Role of Short-Lived Climate Forcers in the Global Climate Regime" (project no.

235548). We thank Nicolas Bellouin for providing RF data for all the models. In addition, we show our appreciation to Nicolas Bellouin, Marianne Tronstad Lund, and Dirk Olivié for giving us vertical distributions of

BC in the Arctic. We thank Daven K. Henze, Forrest Lacey, and three anonymous reviewers for valuable comments that were helpful for the paper. We also thank the editor for his contributions.

Table 1: The models and species included. Models are either general circulation models (GCM) or chemistry transport models (CTM). The resolution column shows the horizontal resolution and the number of vertical layers.

| Model | Type | Resolution | BC | OC | SO$_2$ | NH$_3$ | NO$_x$ | CO | VOC | CH$_4$ | References |
|---|---|---|---|---|---|---|---|---|---|---|---|
| ECHAM6-HAMMOZ | GCM | 1.8°x1.8° L31 | X | X | X | | | | | | Stevens et al. (2013) |
| HadGEM3-GLOMAP | GCM | 1.8°x1.2° L38 | X | X | X | | X | X | X | X | Hewitt et al. (2011) |
| NorESM | GCM | 1.9°x2.5° L26 | X | X | X | | X | X | X | X | Bentsen et al. (2013);Iversen et al. (2013) |
| OsloCTM2 | CTM | 2.8°x2.8° L60 | X | X | X | X | X | X | X | X | Søvde et al. (2008);Myhre et al. (2009) |

[Figure]

[Figure]

Figure 1: ARTP20 for emissions from Europe, East Asia, shipping, and global and for summer and winter. In each frame, and for each emission region, the ARTP20 values for the four latitudinal response bands from south (left) to north (right), as well as the global response average (rightmost), for the species, decomposed by effectsprocesses. The net response is shown by the asterisk. The regions included are Europe (EUR), East Asia (EAS), shipping (SHP), and global (GLB), all for both NH summer, May-October (left), and NH winter, November-April (right). The uncertainty bars show the range across models, which is not given for shipping as the best estimate is based on only two models for that sector. Due to the methodology applied, a fraction of the semi-direct effect for BC in the Arctic is included in the *aerosol effects* process, as explained in Sect. 2.2.4. Note that the vertical axis varies between different emitted components.

[Figure]

Figure 2: ARTP values in different response bands for BC and the ozone precursors for time horizons up to 50 years. Emissions in Europe (left) and East Asia (right) in NH summer (May-October) are given as red and in NH winter (November-April) as blue.

[Figure]

Figure 3: The regional temperature response for a time horizon of 20 years after regional and seasonal emissions in 2008 based
on ARTP(20). The four latitude response bands represent the SH mid-high latitudes, Tropics, NH mid-latitudes, and Arctic.
The global response average is given in Fig. S2. From top to bottom, the emission regions are Europe, East Asia, the global
shipping sector, and global. The emissions are split into NH summer season (May-October) to the left and NH winter season
(November-April) to the right. Note that the y-axis differs for the regions. The horizontal dashed lines show the sum for each
response band.

[Figure]

Figure 4: The relative difference between the global temperature responses based on ARTP and AGTP methods for a time horizon of 20 years. Only cases with larger relative differences than 20% are shown. Positive numbers occur when the magnitude of the global temperature response is larger when based on ARTP than on AGTP, negative when the magnitude is largest based on AGTP.